# Understanding and Improving Fast Adversarial Training

**Maksym Andriushchenko**
EPFL, Theory of Machine Learning Lab
maksym.andriushchenko@epfl.ch

**Nicolas Flammarion**
EPFL, Theory of Machine Learning Lab
nicolas.flammarion@epfl.ch

## Abstract

A recent line of work focused on making adversarial training computationally efficient for deep learning models. In particular, Wong et al. [47] showed that $\ell_\infty$-adversarial training with fast gradient sign method (FGSM) can fail due to a phenomenon called *catastrophic overfitting*, when the model quickly loses its robustness over a single epoch of training. We show that adding a random step to FGSM, as proposed in [47], does not prevent catastrophic overfitting, and that randomness is not important per se — its main role being simply to reduce the magnitude of the perturbation. Moreover, we show that catastrophic overfitting is not inherent to deep and overparametrized networks, but can occur in a single-layer convolutional network with a few filters. In an extreme case, even *a single filter* can make the network highly non-linear *locally*, which is the main reason why FGSM training fails. Based on this observation, we propose a new regularization method, `GradAlign`, that *prevents catastrophic overfitting* by explicitly maximizing the gradient alignment inside the perturbation set and improves the quality of the FGSM solution. As a result, `GradAlign` allows to successfully apply FGSM training also for larger $\ell_\infty$-perturbations and reduce the gap to multi-step adversarial training. The code of our experiments is available at https://github.com/tml-epfl/understanding-fast-adv-training.

## 1 Introduction

Machine learning models based on empirical risk minimization are known to be often non-robust to small worst-case perturbations. For decades, this has been the topic of active research by the statistics, optimization and machine learning communities [19, 2, 10, 3]. However, the recent success of deep learning [22, 33] has raised the interest in this topic. The lack of robustness in deep learning is clearly illustrated by the existence of *adversarial examples*, i.e. tiny input perturbations that can easily fool state-of-the-art deep neural networks into making wrong predictions [38, 12].

Improving the robustness of machine learning models is motivated not only from the security perspective [3]. Adversarially robust models have better interpretability properties [42, 32] and can generalize better [51, 4] including also improved performance under some distribution shifts [48] (although on some performing worse, see [39]). In order to improve the robustness, two families of solutions have been developed: *adversarial training* (AT) that amounts to training the model on adversarial examples [12, 23] and *provable defenses* that derive and optimize robustness certificates [46, 29, 7]. Currently, adversarial-training based methods appear to be preferred by practitioners since they (a) achieve higher empirical robustness (although without providing a robustness certificate), (b) can be scaled to state-of-the-art deep networks without affecting the inference time (unlike smoothing-based approaches [7]), and (c) work equally well for different threat models. Adversarial training can be formulated as a robust optimization problem [35, 23] which takes the form of a non-convex non-concave min-max problem. However, computing the optimal adversarial examples is an NP-hard

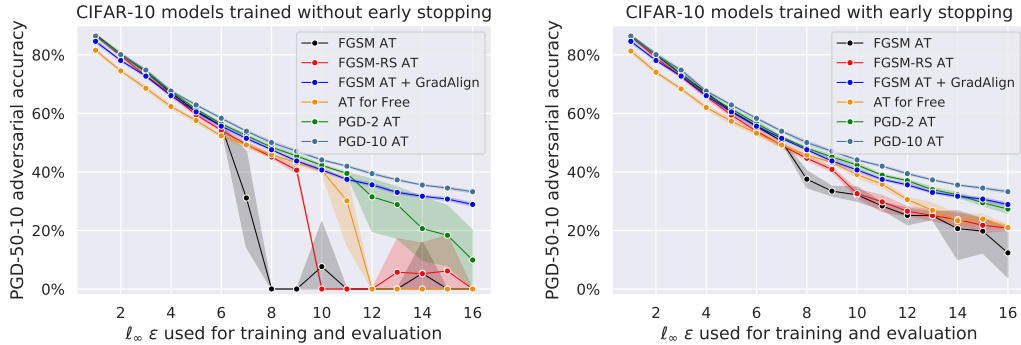

**Figure 1:** Robustness of different adversarial training (AT) methods on CIFAR-10 with ResNet-18 trained and evaluated with different $l_\infty$-radii. The results are averaged over 5 random seeds used for training and reported with the standard deviation. **FGSM AT**: standard FGSM AT, **FGSM-RS AT**: FGSM AT with a random step [47], **FGSM AT + GradAlign**: FGSM AT combined with our proposed regularizer `GradAlign`, **AT for Free**: recently proposed method for fast PGD AT [34], **PGD-2/PGD-10 AT**: AT with a 2-/10-step PGD-attack. Our proposed regularizer `GradAlign` prevents *catastrophic overfitting* in FGSM training and leads to significantly better results which are close to the computationally demanding PGD-10 AT.

problem [21, 45]. Thus adversarial training can only rely on approximate methods to solve the inner maximization problem.

One popular approximation method successfully used in adversarial training is the PGD attack [23] where multiple steps of projected gradient descent are performed. It is now widely believed that models adversarially trained via the PGD attack [23, 50] are robust since small adversarially trained networks can be formally verified [5, 40, 47], and larger models could not be broken on public challenges [23, 50]. Recently, [8] evaluated the majority of recently published defenses to conclude that the standard $\ell_\infty$ PGD training achieves the best empirical robustness; a result which can only be improved using semi-supervised approaches [18, 1, 6]. In contrast, other empirical defenses that were claiming improvements over standard PGD training had overestimated the robustness of their reported models [8]. These experiments imply that adversarial training in general is the key algorithm for robust deep learning, and thus that performing it efficiently is of paramount importance.

Another approximation method for adversarial training is the *Fast Gradient Sign Method* (FGSM) [12] which is based on the linear approximation of the neural network loss function. However, the literature is still ambiguous about the performance of FGSM training, i.e. it remains unclear whether FGSM training can *consistently* lead to robust models. For example, [23] and [41] claim that FGSM training works only for small $\ell_\infty$-perturbations, while [47] suggest that FGSM training can lead to robust models for arbitrary $\ell_\infty$-perturbations if one adds uniformly random initialization before the FGSM step. Related to this, [47] further identified a phenomenon called *catastrophic overfitting* where FGSM training first leads to *some* robustness at the beginning of training, but then suddenly becomes non-robust within a single training epoch. However, the reasons for such a failure remain unknown. This motivates us to consider the following question as the main theme of the paper:

*When and why does fast adversarial training with FGSM lead to robust models?*

**Contributions.** We first show that not only FGSM training is prone to *catastrophic overfitting*, but the recently proposed fast adversarial training methods [34, 47] as well (see Fig. 1). We then analyze the reasons why using a random step in FGSM [47] helps to slightly mitigate catastrophic overfitting and show it simply boils down to reducing the average magnitude of the perturbations. Then we discuss the connection behind catastrophic overfitting and local linearity in deep networks and in single-layer convolutional networks where we show that even *a single filter* can make the network non-linear *locally*, and causes the failure of FGSM training. We additionally provide for this case a theoretical explanation which helps to explain why FGSM AT is successful at the beginning of the training. Finally, we propose a regularization method, `GradAlign`, that *prevents catastrophic overfitting* by explicitly maximizing the gradient alignment inside the perturbation set and therefore improves the quality of the FGSM solution. We compare `GradAlign` to other adversarial training schemes in Fig. 1 and point out that among all fast adversarial training methods considered only

FGSM + `GradAlign` does not suffer from catastrophic overfitting and leads to high robustness even for large $\ell_\infty$-perturbations.

## 2 Problem overview and related work

Let $\ell(x, y; \theta)$ denote the loss of a ReLU-network parametrized by $\theta \in \mathbb{R}^m$ on the example $(x, y) \sim D$ where $D$ is the data generating distribution.[1] Previous works [35, 23] formalized the goal of training adversarially robust models as the following robust optimization problem:

$$\min_\theta \mathbb{E}_{(x,y) \sim D} \left[ \max_{\delta \in \Delta} \ell(x + \delta, y; \theta) \right]. \tag{1}$$

We focus here on the $\ell_\infty$ threat model, i.e. $\Delta = \{\delta \in \mathbb{R}^d, \|\delta\|_\infty \leq \varepsilon\}$, where the adversary can change each input coordinate $x_i$ by at most $\varepsilon$. Unlike classical stochastic saddle point problems of the form $\min_\theta \max_\delta \mathbb{E}[\ell(\theta, \delta)]$ [20], the inner maximization problem here is inside the expectation. Therefore the solution of each subproblem $\max_{\delta \in \Delta} \ell(x + \delta, y; \theta)$ depends on the particular example $(x, y)$ and standard algorithms such as gradient descent-ascent which alternate gradient descent in $\theta$ and gradient ascent in $\delta$ cannot be used. Instead each of these *non-concave* maximization problems has to be solved independently. Thus, an inherent trade-off appears between computationally efficient approaches which aim at solving this inner problem in as few iterations as possible and approaches which aim at solving the problem more accurately but with more iterations. In an extreme case, the PGD attack [23] uses multiple steps of projected gradient ascent (PGD), which is accurate but computationally expensive. At the other end of the spectrum, Fast Gradient Sign Method (FGSM) [12] performs *only* one iteration of gradient ascent with respect to the $\ell_\infty$-norm:

$$\delta_{FGSM} \stackrel{\text{def}}{=} \varepsilon \operatorname{sign}(\nabla_x \ell(x, y; \theta)), \tag{2}$$

followed by a projection of $x + \delta_{FGSM}$ onto the $[0, 1]^d$ to ensure it is a valid input.[2] This leads to a fast algorithm which, however, does not always lead to robust models as observed in [23, 41]. A closer look at the evolution of the robustness during FGSM AT reveals that using FGSM can lead to a model with some degree of robustness but only until a point where the robustness suddenly drops. This phenomenon is called *catastrophic overfitting* in [47]. As a partial solution, the training can be stopped just before that point which leads to non-trivial but suboptimal robustness as illustrated in Fig. 1. Wong et al. [47] further notice that initializing FGSM from a random starting point $\eta \sim \mathcal{U}([-\varepsilon, \varepsilon]^d)$, i.e. using the following perturbation where $\Pi_{[-\varepsilon, \varepsilon]^d}$ denotes the projection:

$$\delta_{FGSM-RS} \stackrel{\text{def}}{=} \Pi_{[-\varepsilon, \varepsilon]^d}[\eta + \alpha \operatorname{sign}(\nabla_x \ell(x + \eta, y; \theta))], \tag{3}$$

helps to mitigate catastrophic overfitting and leads to better robustness for the considered $\varepsilon$ values (e.g. $\varepsilon = 8/255$ on CIFAR-10 in [47]). Along the same lines, [43] observe that using dropout on all layers (including convolutional) also helps to stabilize FGSM AT.

An alternative solution is to interpolate between FGSM and PGD AT. For example, [44] suggest to first use FGSM AT, and later to switch to multi-step PGD AT which is motivated by their analysis suggesting that the inner maximization problem has to be solved more accurately at the end of training. [34] propose to run PGD with step size $\alpha = \varepsilon$ and simultaneously update the weights of the network. On a related note, [49] collect the weight updates during PGD, but apply them after PGD is completed. Additionally, [49] update the gradients of the first layer multiple times. However, none of these approaches are conclusive, either leading to comparable robustness to FGSM-RS training [47] and still failing for higher $\ell_\infty$-radii (see Fig. 1 for [34] and [47]) or being in the worst case as expensive as multi-step PGD AT [44]. Additionally, some previous works deviate from the robust optimization formulation stated in Eq. (1) and instead regularize the model to improve robustness [36, 25, 28], however this does not lead to higher robustness compared to standard adversarial training. We focus next on analyzing the FGSM-RS training [47] as the other recent variations of fast adversarial training [34, 49, 43] lead to models with similar robustness.

**Experimental setup.** Unless mentioned otherwise, we perform training on PreAct ResNet-18 [16] with the cyclic learning rates [37] and half-precision training [24] following the setup of [47]. We evaluate adversarial robustness using the PGD-50-10 attack, i.e. with 50 iterations and 10 restarts with step size $\alpha = \varepsilon/4$ following [47]. More experimental details are specified in Appendix B.

# 3 The role and limitations of using random initialization in FGSM training

First, we show that FGSM with a random step fails to resolve catastrophic overfitting for larger $\varepsilon$. Then we provide evidence against the explanation given by [47] on the benefit of randomness for FGSM AT, and propose a new explanation based on the linear approximation quality of FGSM.

**FGSM with random step does not resolve catastrophic overfitting.** Crucially, [47] observed that adding an initial random step to FGSM as in Eq. (3) helps to avoid catastrophic overfitting. However, this holds only if the step size is not too large (as illustrated in Fig. 3 of [47] for $\varepsilon = 8/255$) and, more importantly, only for small enough $\varepsilon$ as we show in Fig. 1. Indeed, using the step size $\alpha = 1.25\varepsilon$ recommended by [47] *extends* the working regime of FGSM but only from $\varepsilon = 6/255$ to $\varepsilon = 9/255$, with 0% adversarial accuracy for $\varepsilon = 10/255$. When early stopping is applied (Fig. 1, right), there is still a significant gap compared to PGD-10 training, particularly for large $\ell_\infty$-radii. For example, for $\varepsilon = 16/255$, FGSM-RS AT leads to 22.24% PGD-50-10 accuracy while PGD-10 AT obtains a much better accuracy of 30.65%.

**Previous explanation: randomness diversifies the threat model.** A hypothesis stated in [47] was that FGSM-RS helps to avoid catastrophic overfitting by diversifying the threat model. Indeed, the random step allows to have perturbations not only at the corners $\{-\varepsilon, \varepsilon\}^d$ like the FGSM-attack[3], but rather in the whole $\ell_\infty$-ball, $[-\varepsilon, \varepsilon]^d$. Here we refute this hypothesis by modifying the usual PGD training by projecting onto $\{-\varepsilon, \varepsilon\}^d$ the perturbation obtained via the PGD attack. We perform experiments on CIFAR-10 with ResNet-18 with $\ell_\infty$-perturbations of radius $\varepsilon = 8/255$ over 5 random seeds. FGSM AT leads to catastrophic overfitting achieving $0.00 \pm 0.00\%$ adversarial accuracy if early stopping is not applied, while the standard PGD-10 AT and our modified PGD-10 AT schemes achieve $50.48 \pm 0.20\%$ and $50.64 \pm 0.23\%$ adversarial accuracy respectively. Thereby similar robustness as the original PGD AT can still be achieved without training on pertubations from the interior of the $\ell_\infty$-ball. We conclude that *diversity* of adversarial examples is not crucial here. What makes the difference is rather having an iterative instead of a single-step procedure to find a corner of the $\ell_\infty$-ball that sufficiently maximizes the loss.

**New explanation: a random step improves the linear approximation quality.** Using a random step in FGSM is *guaranteed* to decrease the expected magnitude of the perturbation. This simple observation is formalized in the following lemma.

**Lemma 1. (Effect of the random step)** *Let $\eta \sim \mathcal{U}([-\varepsilon, \varepsilon]^d)$ be a random starting point, and $\alpha \in [0, 2\varepsilon]$ be the step size of FGSM-RS defined in Eq. (3), then*

$$\mathbb{E}_\eta\left[\|\delta_{FGSM-RS}(\eta)\|_2\right] \leq \sqrt{\mathbb{E}_\eta\left[\|\delta_{FGSM-RS}(\eta)\|_2^2\right]} = \sqrt{d}\sqrt{-\frac{1}{6\varepsilon}\alpha^3 + \frac{1}{2}\alpha^2 + \frac{1}{3}\varepsilon^2}. \quad (4)$$

The proof is deferred to Appendix A.1. We first remark that the upper bound is in the range $[1/\sqrt{3}\sqrt{d}\varepsilon, \sqrt{d}\varepsilon]$, and therefore always less or equal than $\|\delta_{FGSM}\|_2 = \sqrt{d}\varepsilon$. We visualize our bound in Fig. 2 where the expectation is approximated by Monte-Carlo sampling over 1,000 samples of $\eta$, and note that the bound becomes increasingly tight for high-dimensional inputs.

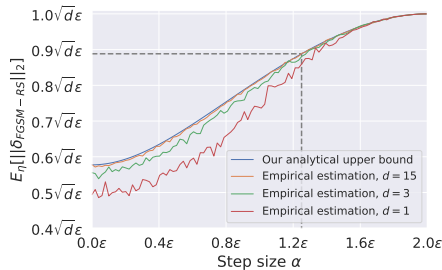

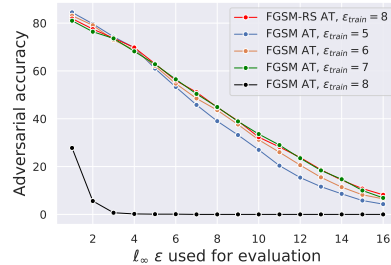

**Figure 2:** Visualization of our upper bound on $\mathbb{E}_\eta[\|\delta_{FGSM-RS}\|_2]$. The dashed line corresponds to the step size $\alpha = 1.25\varepsilon$ recommended in [47].

**Figure 3:** Robustness of FGSM-trained ResNet-18 on CIFAR-10 with different $\varepsilon_{train}$ used for training compared to FGSM-RS AT with $\varepsilon_{train} = 8/255$.

**Table 1:** Robustness of FGSM AT with a reduced step size ($\alpha = \frac{7}{255}$) compared to the FGSM-RS AT proposed in [47] ($\alpha = \frac{10}{255}$) for $\varepsilon = \frac{8}{255}$ on CIFAR-10 for ResNet-18 trained with early stopping. The results are averaged over 5 random seeds used for training.

| Model<br>Accuracy | FGSM AT | FGSM $\alpha = \frac{7}{255}$ AT | FGSM-RS AT |
|---|---|---|---|
| PGD-50-10 | $36.35 \pm 1.74\%$ | $45.35 \pm 0.48\%$ | $45.60 \pm 0.19\%$ |

The key observation here is that among all possible perturbations of $\ell_\infty$-norm $\varepsilon$, perturbations with a smaller $\ell_2$-norm benefit from a better linear approximation. This statement follows from the second-order Taylor expansion for twice differentiable functions:

$$f(x + \delta) \approx f(x) + \langle \nabla_x f(x), \delta \rangle + \langle \delta, \nabla^2_{xx} f(x)\delta \rangle,$$

i.e. a smaller value of $\|\delta\|_2^2$ implies a smaller linear approximation error $|f(x + \delta) - f(x) - \langle \nabla_x f(x), \delta \rangle |$. Moreover, the same property still holds empirically for the non-differentiable ReLU networks (see Appendix C.1). We conclude that by reducing in expectation the length of the perturbation $\|\delta\|_2$, the FGSM-RS approach of [47] takes advantage of a better linear approximation. This is supported by the fact that FGSM-RS AT also leads to catastrophic overfitting if the step size $\alpha$ is chosen to be too large (see Fig. 3 in [47]), thus providing no benefits over FGSM AT even when combined with early stopping. We argue this is the main improvement over the standard FGSM AT.

**Successful FGSM AT does not require randomness.** If having perturbation with a too large $\ell_2$-norm is indeed the key factor in catastrophic overfitting, we can expect that just reducing the step size of the standard FGSM should work equally well as FGSM-RS. For $\varepsilon = \frac{8}{255}$ on CIFAR-10, [47] recommend to use FGSM-RS with step size $\alpha = 1.25\varepsilon$ which induces a perturbation of expected $\ell_2$-norm $\|\delta_{FGSM-RS}\|_2 \approx \frac{7}{255}\sqrt{d}$. This corresponds to using standard FGSM with a step size $\alpha \approx \frac{7}{255}$ instead of $\alpha = \varepsilon = \frac{8}{255}$ (see the dashed line in Fig. 2). We report the results in Table 1 and observe that simply reducing the step size of FGSM (without *any* randomness) leads to the same level of robustness. We show further in Fig. 3 that when used with a smaller step size, the robustness of *standard* FGSM training even without early stopping can generalize to much higher $\varepsilon$. This contrasts with the previous literature [23, 41]. We conclude from these experiments that a more direct way to improve FGSM AT and to prevent it from catastrophic overfitting is to simply reduce the step size. Note that this still leads to suboptimal robustness compared to PGD AT (see Fig. 1) for $\varepsilon$ larger than the one used during training, since in this case adversarial examples can only be generated inside the smaller $\ell_\infty$-ball. This motivates us to take a closer look on *how* and *why* catastrophic overfitting occurs to be able to prevent it without reducing the FGSM step size.

## 4 Understanding catastrophic overfitting via gradient alignment

First, we establish a connection between catastrophic overfitting and local linearity of the model. Then we show that catastrophic overfitting also occurs in a single-layer convolutional network, for which we analyze local linearity both empirically and theoretically.

**When can the inner maximization problem be accurately solved with FGSM?** Recall that the FGSM attack [12] is obtained as a closed-form solution of the following optimization problem: $\delta_{FGSM} = \arg\max_{\|\delta\|_\infty \leq \varepsilon} \langle \nabla_x \ell(x, y; \theta)), \delta \rangle$. Thus, the FGSM attack is guaranteed to find the optimal adversarial perturbation if $\nabla_x \ell(x, y; \theta)$ is constant inside the $\ell_\infty$-ball around the input $x$, i.e. the loss function is *locally linear*. This motivates us to study the evolution of local linearity during FGSM training and its connection to catastrophic overfitting. With this aim, we define the following local linearity metric of the loss function $\ell$:

$$\mathbb{E}_{(x,y)\sim D,\ \eta\sim\mathcal{U}([-\varepsilon,\varepsilon]^d)} \left[ \cos\left( \nabla_x \ell(x, y; \theta), \nabla_x \ell(x + \eta, y; \theta) \right) \right], \tag{5}$$

which we refer to as *gradient alignment*. This quantity is easily interpretable: it is equal to one for models linear inside the $\ell_\infty$-ball of radius $\varepsilon$, and it is approximately zero when the input gradients are nearly orthogonal to each other. Previous works also considered local linearity of deep networks [25, 28], however rather with the goal of introducing regularization methods that improve robustness as an *alternative* to adversarial training. More precisely, [25] propose to use a curvature regularization

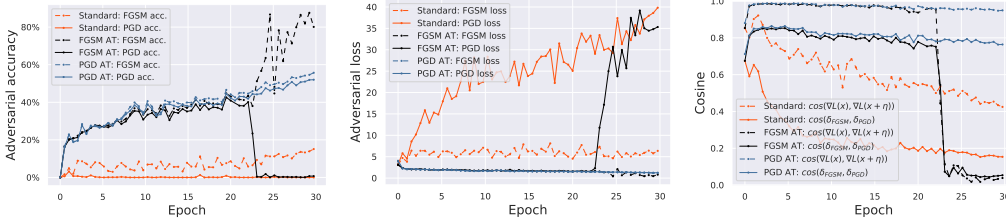

**Figure 4:** Visualization of the training process of standardly trained, FGSM trained, and PGD-10 trained ResNet-18 on CIFAR-10 with $\varepsilon = {}^8/_{255}$. All the statistics are calculated on the test set. Catastrophic overfitting for the FGSM AT model occurs around epoch 23 and is characterized by a sudden drop in the PGD accuracy, a gap between the FGSM and PGD losses, and a dramatic decrease of *local linearity*.

method that uses the FGSM point, and [28] find the input point where local linearity is maximally violated using an iterative method, leading to comparable computational cost as PGD AT. In contrast, we analyze here gradient alignment to improve FGSM training without seeking an alternative to it.

**Catastrophic overfitting in deep networks.** To understand the link between catastrophic overfitting and local linearity, we plot in Fig. 4 the adversarial accuracies and the loss values obtained by FGSM and PGD AT on CIFAR-10 using ResNet-18, together with the gradient alignment (see Eq. 5) and the cosine between FGSM and PGD perturbations. We compute these statistics on the test set. Catastrophic overfitting occurs for FGSM AT around epoch 23, and is characterized by the following intertwined events: (a) There is a *sudden drop* in the PGD accuracy from $40.1\%$ to $0.0\%$, along with an *abrupt jump* of the FGSM accuracy from $43.5\%$ to $86.7\%$. In contrast, before the catastrophic overfitting, the ratio between the average PGD and FGSM losses never exceeded $1.05$. This suggests that FGSM cannot anymore accurately solve the inner maximization problem. (b) Concurrently, after catastrophic overfitting, the gradient alignment of the FGSM model experiences a *phase transition* and drops significantly from $0.95$ to $0.05$ within an epoch of training, i.e. *the input gradients become nearly orthogonal inside the $\ell_\infty$-ball*. We observe the same drop also for $\cos(\delta_{FGSM}, \delta_{PGD})$ which means that the FGSM and PGD directions are not aligned anymore (as also observed in [41]). This echoes the observation made in [26] that SGD on the standard loss of a neural network learns models of increasing complexity. We observe qualitatively the same phenomenon for FGSM AT, where the complexity is captured by the degree of local non-linearity. The connection between local linearity and catastrophic overfitting sparks interest for a further analysis in a simpler setting.

**Catastrophic overfitting in a single-layer CNN.** We show that catastrophic overfitting is not inherent to deep and overparametrized networks, and can be observed in a very simple setup. For this we train a single-layer CNN with four filters on CIFAR-10 using FGSM AT with $\varepsilon = {}^{10}/_{255}$ (see Sec. B for details). We observe that catastrophic overfitting occurs in this simple model as well, and its pattern is the same as in ResNet: a simultaneous drop of the PGD accuracy and gradient alignment (see Appendix C.2). The advantage of considering a simple model is that we can inspect the learned filters and understand what causes the network to become highly non-linear locally. We observe that after catastrophic overfitting the network has learned in filter $w_4$ a variant of the Laplace filter (see Fig. 5), an edge-detector filter which is well-known for *amplifying high-frequency noise* such as uniform noise [11]. Until the end of training, filter $w_4$ preserves its direction (see Appendix C.2 for detailed visualizations), but grows significantly in its magnitude together with its outcoming weights, in contrast to the rest of the filters as shown in Fig. 5. Interestingly, if we set $w_4$ to zero, the network largely *recovers local linearity*: the gradient alignment increases from $0.08$ to $0.71$, recovering its value before catastrophic overfitting. Thus, in this extreme case, *even a single convolutional filter can cause catastrophic overfitting*. Next we analyze formally gradient alignment in a single-layer CNN and elaborate on the connection to the noise sensitivity.

**Analysis of gradient alignment in a single-layer CNN.** We analyze here a single-layer CNN with ReLU-activation. Let $Z \in \mathbb{R}^{p \times k}$ be the matrix of $k$ non-overlapping image patches extracted from the image $x = \text{vec}(Z) \in \mathbb{R}^d$ such that $z_j = z_j(x) \in \mathbb{R}^p$. The model prediction $f$ is parametrized by $(W, b, U, c) \in \mathbb{R}^{p \times m} \times \mathbb{R}^m \times \mathbb{R}^{m \times k} \times \mathbb{R}$, and its prediction and the input gradient are given as

$$f(x) = \sum_{i=1}^{m} \sum_{j=1}^{k} u_{ij} \max\{\langle w_i, z_j \rangle + b_i, 0\} + c, \quad \nabla_x f(x) = \text{vec}\left(\sum_{i=1}^{m} \sum_{j=1}^{k} u_{ij} \mathbb{1}_{\langle w_i, z_j \rangle + b_i \geq 0} w_i e_j^T\right).$$

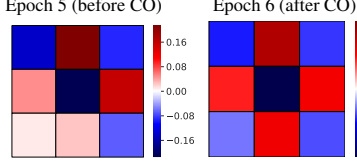

Epoch 5 (before CO)  Epoch 6 (after CO)

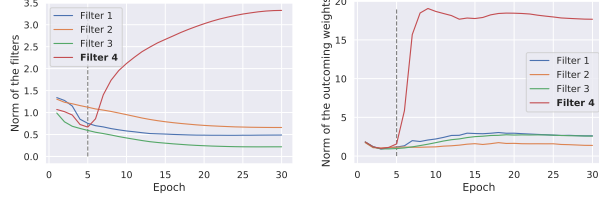

**Figure 5:** Filter $w_4$ (green channel) in a single-layer CNN before and after catastrophic overfitting (CO).

**Figure 6:** Evolution of the weight norms in a single-layer CNN before and after catastrophic overfitting (dashed line).

We observe that catastrophic overfitting only happens at later stages of training. At the beginnning of the training, the gradient alignment is very high (see Fig. 4 and Fig. 11), and FGSM solves the inner maximization problem accurately enough. Thus, an important aspect of FGSM training is that the model starts training from *highly aligned gradient*. This motivates us to inspect closely gradient alignment at initialization.

**Lemma 2. (Gradient alignment at initialization)** *Let $z \sim \mathcal{U}([0,1]^p)$ be an image patch for $p \geq 2$, $\eta \sim \mathcal{U}([-\varepsilon, \varepsilon]^d)$ a point inside the $\ell_\infty$-ball, the parameters of a single-layer CNN initialized i.i.d. as $w \sim \mathcal{N}(0, \sigma_w^2 I_p)$ for every column of $W$, $u \sim \mathcal{N}(0, \sigma_u^2 I_m)$ for every column of $U$, $b := 0$, then the gradient alignment is lower bounded by*

$$\lim_{k,m \to \infty} \cos\left(\nabla_x \ell(x,y), \nabla_x \ell(x+\eta, y)\right) \geq \max\left\{1 - \sqrt{2}\, \mathbb{E}_{w,z}\left[e^{-\frac{1}{\varepsilon^2}\langle w/\|w\|_2, z\rangle^2}\right]^{1/2}, 0.5\right\}.$$

The lemma implies that for randomly initialized CNNs with a large enough number of image patches $k$ and filters $m$, gradient alignment cannot be smaller than $0.5$. This is in contrast to the value of $0.12$ that we observe after catastrophic overfitting when the weights are no longer i.i.d. We note that the lower bound of $0.5$ is quite pessimistic since it holds for an arbitrarily large $\varepsilon$. The lower bound is close to $1$ when $\varepsilon$ is small compared to $\mathbb{E}\|z\|_2$ which is typical in adversarial robustness (see Appendix A.2 for the visualization of the lower bound). High gradient alignment at initialization also holds empirically for deep networks as well, e.g. for ResNet-18 (see Fig. 4), starting from the value of $0.85$ in contrast to $0.04$ after catastrophic overfitting. Thus, it appears to be a general phenomenon that the standard

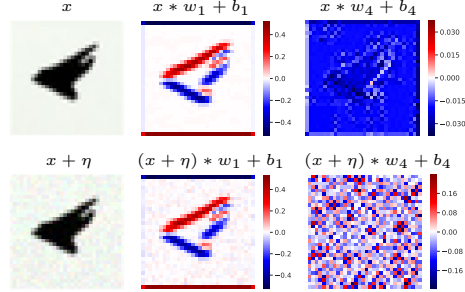

**Figure 7:** Feature maps of filters $w_1$ and $w_4$ in a single-layer CNN. A small noise $\eta$ is significantly amplified by the Laplace filter $w_4$ in contrast to a regular filter $w_1$.

initialization scheme of neural network weights [15] ensures the *initial* success of FGSM training.

In contrast, after some point during training, the network can learn parameters which lead to a significant reduction of gradient alignment. For simplicity, let us consider a single-filter CNN where the gradient alignment for a filter $w$ and bias $b$ at points $x$ and $x + \eta$ has a simple expression:

$$\cos\left(\nabla_x \ell(x,y), \nabla_x \ell(x+\eta, y)\right) = \frac{\sum_{i=1}^k u_i^2 \mathbb{1}_{\langle w, z_i\rangle + b \geq 0} \mathbb{1}_{\langle w, z_i+\eta_i\rangle + b \geq 0}}{\sqrt{\sum_{i=1}^k u_i^2 \mathbb{1}_{\langle w, z_i\rangle + b \geq 0} \sum_{i=1}^k u_i^2 \mathbb{1}_{\langle w, z_i+\eta_i\rangle + b \geq 0}}}. \tag{6}$$

Considering a single-filter CNN is also motivated by the fact that in the single-layer CNN introduced earlier, the norms of $w_4$ and its outcoming weights are much higher than for the rest of the filters (see Fig. 6), and thus the contribution of $w_4$ to the predictions and gradients of the network is the most significant. We observe that when an image $x$ is convolved with the Laplace filter $w_4$, even a uniformly random noise $\eta$ of small magnitude is able to significantly affect the output of $(x+\eta) * w_4$ (see Fig. 7). As a consequence, the ReLU activations of the network change their signs which directly affects the gradient alignment in Eq. (6). Namely, $x * w_4 + b_4$ has mostly negative values, and thus many values $\{\mathbb{1}_{\langle w_4, z_i\rangle + b_4}\}_{i=1}^k$ are equal to 0. On the other hand, nearly half of the values $\{\mathbb{1}_{\langle w_4, z_i+\eta_i\rangle + b_4}\}_{i=1}^k$ become 1, which significantly increases the denominator of Eq. (6), and thus makes the cosine close to 0. At the same time, the output of a regular filter $w_1$ shown in Fig. 7 is only slightly affected by the random noise $\eta$. For deep networks, however, we could not identify *particular* filters responsible for catastrophic overfitting, thus we consider next a more general solution.

# 5 Increasing gradient alignment improves fast adversarial training

Based on the importance of gradient alignment for successful FGSM training, we propose a regularizer, `GradAlign`, that aims at increasing gradient alignment and preventing catastrophic overfitting. The core idea of `GradAlign` is to maximize the gradient alignment (as defined in Eq. (5)) between the gradients at point $x$ and at a randomly perturbed point $x + \eta$ inside the $\ell_\infty$-ball around $x$:

$$\Omega(x, y, \theta) \stackrel{\text{def}}{=} \mathbb{E}_{(x,y) \sim D, \, \eta \sim \mathcal{U}([-\varepsilon, \varepsilon]^d)} \left[ 1 - \cos \left( \nabla_x \ell(x, y; \theta), \nabla_x \ell(x + \eta, y; \theta) \right) \right]. \tag{7}$$

Crucially, `GradAlign` uses gradients at points $x$ and $x + \eta$ which does not require an expensive iterative procedure unlike, e.g., the LLR method of [28]. Note that the regularizer depends only on the gradient direction and it is invariant to the gradient norm which contrasts it to the gradient penalties [14, 17, 31, 36] or CURE [25] (see the comparison in Appendix D.5).

**Experimental setup.** We compare the following methods: standard FGSM AT, FGSM-RS AT with $\alpha = 1.25\varepsilon$ [47], FGSM AT + `GradAlign`, *AT for Free* with $m = 8$ [34], PGD-2 AT with 2-step PGD using $\alpha = \varepsilon/2$, and PGD-10 AT with 10-step PGD using $\alpha = 2\varepsilon/10$. We train these methods using PreAct ResNet-18 [16] with $\ell_\infty$-radii $\varepsilon \in \{1/255, \dots, 16/255\}$ on CIFAR-10 for 30 epochs and $\varepsilon \in \{1/255, \dots, 12/255\}$ on SVHN for 15 epochs. The only exception is *AT for Free* [34] which we train for 96 epochs on CIFAR-10, and 45 epochs on SVHN which was necessary to get comparable results to the other methods. Unlike [28] and [49], with the training scheme of [47] and $\alpha = \varepsilon/2$ we could successfully train a PGD-2 model with $\varepsilon = 8/255$ on CIFAR-10 with robustness better than that of their methods that use the same number of PGD steps (see Appendix D.3). This also echoes the recent finding of [30] that properly tuned multi-step PGD AT outperforms more recently published methods. As before, we evaluate robustness using PGD-50-10 with 50 iterations and 10 restarts using step size $\alpha = \varepsilon/4$ following [47] for the same $\varepsilon$ that was used for training. We train each model with 5 random seeds since the final robustness can have a large variance for high $\varepsilon$. Also, we remark that training with `GradAlign` leads on average to a $3\times$ slowdown on an NVIDA V100 GPU compared to FGSM training which is due to the use of double backpropagation (see [9] for a detailed analysis). We think that improving the runtime of `GradAlign` is possible, but we postpone it to future work. Additional implementation details are provided in Appendix B. The code of our experiments is available at https://github.com/tml-epfl/understanding-fast-adv-training.

**Results on CIFAR-10 and SVHN.** We provide the main comparison in Fig. 8 and provide detailed numbers for specific values of $\varepsilon$ in Appendix D.3 which also includes an additional evaluation of our models with *AutoAttack* [8]. First, we notice that all the methods perform almost equally well for small enough $\varepsilon$, i.e. $\varepsilon \leq 6/255$ on CIFAR-10 and $\varepsilon \leq 4/255$ on SVHN. However, the performance for larger $\varepsilon$ varies a lot depending on the method due to catastrophic overfitting. Importantly, `GradAlign` *succesfully prevents catastrophic overfitting* in FGSM AT, thus allowing to successfully apply FGSM training also for larger $\ell_\infty$-perturbations and reduce the gap to PGD-10 training. In Appendix D.4, we additionally show that FGSM + `GradAlign` does not suffer from catastrophic overfitting even for $\varepsilon \in \{24/255, 32/255\}$. At the same time, *not only* FGSM AT and FGSM-RS AT experience catastrophic overfitting, but also the recently proposed *AT for Free* and PGD-2, although at higher $\varepsilon$ values than

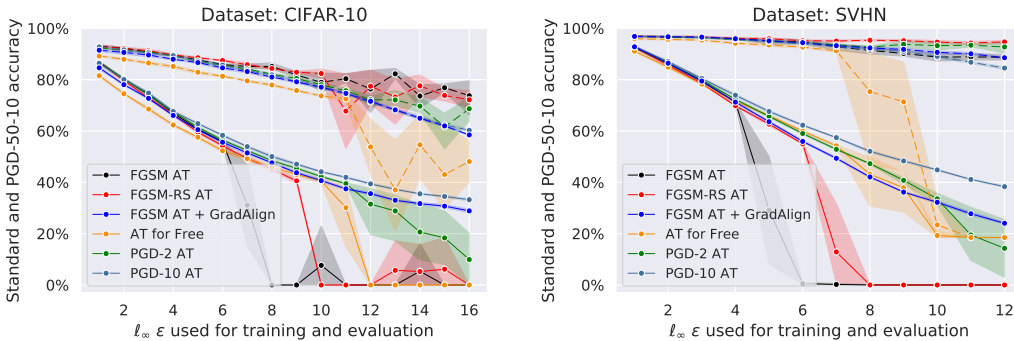

**Figure 8:** Accuracy (dashed line) and robustness (solid line) of different adversarial training (AT) methods on CIFAR-10 and SVHN with ResNet-18 trained and evaluated with different $l_\infty$-radii. The results are obtained without early stopping, averaged over 5 random seeds used for training and reported with the standard deviation.

FGSM AT. We note that `GradAlign` is not only applicable to FGSM AT, but also to other methods that can also suffer from catastrophic overfitting. In particular, combining PGD-2 with `GradAlign` prevents catastrophic overfitting and leads to better robustness for $\varepsilon = {}^{16}/{255}$ on CIFAR-10 (see Appendix D.3). Although performing early stopping can lead to non-trivial robustness, standard accuracy is often significantly sacrificed which *limits the usefulness of early stopping* as we show in Appendix D.2. This is in contrast to training with `GradAlign` which leads to the same standard accuracy as PGD-10 AT.

**Results on ImageNet.** We also performed similar experiments on ImageNet in Appendix D.3 to illustrate that `GradAlign` can be scaled to large-scale problems despite the slowdown. However, we observed that even for standard FGSM training using the training schedule of [47], catastrophic overfitting *does not* occur for $\varepsilon \in \{{}^2/{255}, {}^4/{255}\}$ considered in [34, 47], and thus there is no need to use `GradAlign` as its main role is to prevent catastrophic overfitting. We observe that for these $\varepsilon$ values, the gradient alignment evolves similarly to that of PGD AT from the CIFAR-10 experiments shown in Fig. 4, i.e. it decreases gradually over epochs but *without* a sharp drop that would indicate catastrophic overfitting. For $\varepsilon = {}^6/{255}$, we observe that the gradient alignment and PGD accuracy for FGSM-RS drop very early in training (after 3 epochs), but not for FGSM or FGSM + `GradAlign` training. This contradicts our observations on CIFAR-10 and SVHN where we observed that FGSM-RS usually helps to postpone catastrophic overfitting to higher $\varepsilon$. However, it is computationally demanding to replicate the results on ImageNet over different random seeds as we did for CIFAR-10 and SVHN. Thus, we leave a more detailed investigation of catastrophic overfitting on ImageNet for future work.

**Robust vs. catastrophic overfitting.** Recently, Rice et al. [30] brought up the importance of early stopping in adversarial training to prevent the *robust overfitting* phenomenon when training longer hurts the adversarial accuracy on the test set. It is thus a natural question to ask whether robust and catastrophic overfitting are related, and whether `GradAlign` can be beneficial to mitigate robust overfitting. We observed that training FGSM + `GradAlign` for more than 30 epochs also leads to slightly worse robustness on the test set (see Appendix D.4), thus suggesting that *catastrophic* and *robust overfitting* are two distinct phenomena that have to be addressed separately. As a sidenote, we also observe that FGSM training combined with `GradAlign` does not lead to catastrophic overfitting even when trained *up to 200 epochs*.

# 6 Conclusions and outlook

We observed that catastrophic overfitting is a fundamental problem not only for standard FGSM training, but for computationally efficient adversarial training in general. In particular, many recently proposed schemes such as FGSM AT enhanced by a random step or *AT for free* are also prone to catastrophic overfitting, and early stopping leads to suboptimal models. Motivated by this, we explored the questions of *when* and *why* FGSM adversarial training works, and how to improve it by increasing the gradient alignment, and thus the quality of the solution of the inner maximization problem. Our proposed regularizer, `GradAlign`, prevents catastrophic overfitting and improves the robustness compared to other fast adversarial training methods reducing the gap to multi-step PGD training. However, `GradAlign` leads to an increased runtime due to the use of double backpropagation. We hope that the same effect of stabilizing the gradients under *random* noise can be achieved in future work with other regularization methods that do not rely on double backpropagation.

## Acknowledgments and Disclosure of Funding

We thank Eric Wong, Francesco Croce, and Chen Liu for many fruitful discussions. We are also very grateful to Guillermo Ortiz-Jimenez, Apostolos Modas, Ludwig Schmidt, and anonymous reviewers for the useful feedback on the paper.

## Broader Impact

Our work focuses on a systematic study of the failure reasons behind computationally efficient adversarial training methods. We suggest a new regularization approach which helps to overcome the shortcoming of the existing methods that is known as *catastrophic overfitting*.

We see primarily positive outcomes from our work since adversarial robustness is a desirable property that improves the *reliability* of machine learning models. Therefore, it is crucial to be able to train robust models efficiently and without limiting efficient training only to perturbations of a small size.

## Footnotes

[1] In practice we use training samples with random data augmentation.

[2] Throughout the paper we will focus on image classification, i.e. inputs $x$ will be images.

[3]For simplicity, we ignore the projection of $x + \delta$ onto $[0, 1]^d$ in this section.

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
