[Supplementary Material]

# Appendix

## A    Deferred proofs

In this section, we show the proofs omitted from Sec. 3 and Sec. 4.

### A.1    Proof of Lemma 1

We state again Lemma 1 from Sec. 3 and present the proof.

**Lemma 1. (Effect of the random step)** *Let $\eta \sim \mathcal{U}([-\varepsilon, \varepsilon]^d)$ be a random starting point, and $\alpha \in [0, 2\varepsilon]$ be the step size of FGSM-RS defined in Eq. (3), then*

$$\mathbb{E}_\eta \left[\|\delta_{FGSM-RS}(\eta)\|_2\right] \leq \sqrt{\mathbb{E}_\eta \left[\|\delta_{FGSM-RS}(\eta)\|_2^2\right]} = \sqrt{d}\sqrt{-\frac{1}{6\varepsilon}\alpha^3 + \frac{1}{2}\alpha^2 + \frac{1}{3}\varepsilon^2}.$$

*Proof.* First, note that due to the Jensen's inequality, we can have a convenient upper bound which is easier to work with:

$$\mathbb{E}\left[\|\delta_{FGSM-RS}(\eta)\|_2\right] \leq \sqrt{\mathbb{E}\left[\|\delta_{FGSM-RS}(\eta)\|_2^2\right]}. \tag{8}$$

Therefore, we can focus on $\mathbb{E}\left[\|\delta_{FGSM-RS}\|_2^2\right]$ which can be computed analytically. Let us denote by $\nabla \overset{\text{def}}{=} \nabla_x \ell(x + \eta, y; \theta) \in \mathbb{R}^d$, we then obtain:

$$\mathbb{E}_\eta \left[\|\delta_{FGSM-RS}\|_2^2\right] = \mathbb{E}_\eta \left[\|\Pi_{[-\varepsilon,\varepsilon]}\left[\eta + \alpha \operatorname{sign}(\nabla)\right]\|_2^2\right] = \sum_{i=1}^d \mathbb{E}_{\eta_i}\left[\Pi_{[-\varepsilon,\varepsilon]}\left[\eta_i + \alpha \operatorname{sign}(\nabla_i)\right]^2\right]$$

$$= d\,\mathbb{E}_{\eta_i}\left[\min\{\varepsilon, |\eta_i + \alpha \operatorname{sign}(\nabla_i)|\}^2\right] = d\,\mathbb{E}_{\eta_i}\left[\min\{\varepsilon^2, (\eta_i + \alpha \operatorname{sign}(\nabla_i))^2\}\right]$$

$$= d\,\mathbb{E}_{r_i}\left[\mathbb{E}_{\eta_i}\left[\min\{\varepsilon^2, (\eta_i + \alpha \operatorname{sign}(\nabla_i))^2\} \mid \operatorname{sign}(\nabla_i) = r_i\right]\right],$$

where in the last step we use the law of total expectation by noting that $\operatorname{sign}(\nabla_i)$ is also a random variable since it depends on $\eta_i$.

We first consider the case when $\operatorname{sign}(\nabla_i) = 1$, then the inner conditional expectation is equal to:

$$\int_{-\varepsilon}^{\varepsilon} \min\{\varepsilon^2, (\eta_i + \alpha)^2\} \frac{1}{2\varepsilon} d\eta_i = \frac{1}{2\varepsilon} \int_{-\varepsilon+\alpha}^{\varepsilon+\alpha} \min\{\varepsilon^2, x^2\} dx$$

$$= \frac{1}{2\varepsilon}\left(\int_{\varepsilon}^{\varepsilon+\alpha} \varepsilon^2 dx + \int_{-\varepsilon+\alpha}^{\varepsilon} x^2 dx\right)$$

$$= -\frac{1}{6\varepsilon}\alpha^3 + \frac{1}{2}\alpha^2 + \frac{1}{3}\varepsilon^2.$$

The case when $\operatorname{sign}(\nabla_i) = -1$ leads to the same expression:

$$\int_{-\varepsilon}^{\varepsilon} \min\{\varepsilon^2, (\eta_i - \alpha)^2\} \frac{1}{2\varepsilon} d\eta_i = \frac{1}{2\varepsilon} \int_{-\varepsilon-\alpha}^{\varepsilon-\alpha} \min\{\varepsilon^2, x^2\} dx = -\frac{1}{6\varepsilon}\alpha^3 + \frac{1}{2}\alpha^2 + \frac{1}{3}\varepsilon^2.$$

Combining these two cases together with Eq. (8), we have that:

$$\mathbb{E}_\eta \left[\|\delta_{FGSM-RS}(\eta)\|_2\right] \leq \sqrt{\mathbb{E}\left[\|\delta_{FGSM-RS}(\eta)\|_2^2\right]} = \sqrt{d}\sqrt{-\frac{1}{6\varepsilon}\alpha^3 + \frac{1}{2}\alpha^2 + \frac{1}{3}\varepsilon^2}.$$

$$\square$$

## A.2 Proof and discussion of Lemma 2

We state again Lemma 2 from Sec. 4 and present the proof.

**Lemma 2. (Gradient alignment at initialization)** *Let $z \sim \mathcal{U}([0,1]^p)$ be an image patch for $p \geq 2$, $\eta \sim \mathcal{U}([-\varepsilon, \varepsilon]^d)$ a point inside the $\ell_\infty$-ball, the parameters of a single-layer CNN initialized i.i.d. as $w \sim \mathcal{N}(0, \sigma_w^2 I_p)$ for every column of $W$, $u \sim \mathcal{N}(0, \sigma_u^2 I_m)$ for every column of $U$, $b := 0$, then the gradient alignment is lower bounded by*

$$\lim_{k,m \to \infty} \cos\left(\nabla_x \ell(x,y), \nabla_x \ell(x+\eta, y)\right) \geq \max\left\{1 - \sqrt{2}\, \mathbb{E}_{w,z}\left[e^{-\frac{1}{\varepsilon^2}\langle w/\|w\|_2, z\rangle^2}\right]^{1/2}, 0.5\right\}.$$

*Proof.* For $k$ and $m$ large enough, the law of large number ensures that an empirical mean of i.i.d. random variables can be approximated by its expectation with respect to random variables $z, \eta, w, u$. This leads to

$$\lim_{k,m \to \infty} \cos\left(\nabla_x \ell(x,y), \nabla_x \ell(x+\eta, y)\right)$$

$$= \lim_{k,m \to \infty} \frac{\sum_{r=1}^{m}\sum_{l=1}^{m}\sum_{i=1}^{k} \langle w_r, w_l \rangle\, u_{ri} u_{li} \mathbb{1}_{\langle w_r, z_i\rangle \geq 0} \mathbb{1}_{\langle w_l, z_i+\eta_i\rangle \geq 0}}{\sqrt{\sum_{r=1}^{m}\sum_{l=1}^{m}\sum_{i=1}^{k} \langle w_r, w_l\rangle\, u_{ri} u_{li} \mathbb{1}_{\langle w_r, z_i\rangle \geq 0}\mathbb{1}_{\langle w_l, z_i\rangle \geq 0}} \sqrt{\sum_{r=1}^{m}\sum_{l=1}^{m}\sum_{i=1}^{k} \langle w_r, w_l\rangle\, u_{ri} u_{li} \mathbb{1}_{\langle w_r, z_i+\eta_i\rangle \geq 0}\mathbb{1}_{\langle w_l, z_i+\eta_i\rangle \geq 0}}}$$

$$= \lim_{k,m \to \infty} \frac{\frac{1}{km}\sum_{r=1}^{m}\sum_{i=1}^{k} \|w_r\|_2^2\, u_{ri}^2 \mathbb{1}_{\langle w_r, z_i\rangle \geq 0}\mathbb{1}_{\langle w_r, z_i+\eta_i\rangle \geq 0}}{\sqrt{\frac{1}{km}\sum_{r=1}^{m}\sum_{i=1}^{k} \|w_r\|_2^2\, u_{ri}^2 \mathbb{1}_{\langle w_r, z_i\rangle \geq 0}\mathbb{1}_{\langle w_r, z_i\rangle \geq 0}} \sqrt{\frac{1}{km}\sum_{r=1}^{m}\sum_{i=1}^{k} \|w_r\|_2^2\, u_{ri}^2 \mathbb{1}_{\langle w_r, z_i+\eta_i\rangle \geq 0}\mathbb{1}_{\langle w_r, z_i+\eta_i\rangle \geq 0}}}$$

$$= \frac{\mathbb{E}_{w,u,\eta,z}\left[\|w\|_2^2\, u^2 \mathbb{1}_{\langle w,z\rangle \geq 0}\mathbb{1}_{\langle w,z+\eta\rangle \geq 0}\right]}{\sqrt{\mathbb{E}_{w,u,z}\left[\|w\|_2^2\, u^2 \mathbb{1}_{\langle w,z\rangle \geq 0}\right]}\sqrt{\mathbb{E}_{w,u,\eta,z}\left[\|w\|_2^2\, u^2 \mathbb{1}_{\langle w,z+\eta\rangle \geq 0}\right]}}$$

$$= \frac{\mathbb{E}_{w,z,\eta}\left[\|w\|_2^2\, \mathbb{1}_{\langle w,z\rangle \geq 0}\mathbb{1}_{\langle w,z+\eta\rangle \geq 0}\right]}{\sqrt{\mathbb{E}_{w,z}\left[\|w\|_2^2\, \mathbb{1}_{\langle w,z\rangle \geq 0}\right]}\sqrt{\mathbb{E}_{w,z,\eta}\left[\|w\|_2^2\, \mathbb{1}_{\langle w,z+\eta\rangle \geq 0}\right]}}. \tag{9}$$

We directly compute for the denominator:

$$\mathbb{E}_{w,z}[\|w\|_2^2\, \mathbb{1}_{\langle w,z\rangle \geq 0}] = \mathbb{E}_{w,\eta,z}[\|w\|_2^2\, \mathbb{1}_{\langle w,z+\eta\rangle \geq 0}] = 0.5 p \sigma_w^2.$$

For the numerator, by bounding $\mathbb{P}_\eta\left[\langle w, \eta\rangle \geq \langle w, z\rangle\right] \leq e^{-\frac{\langle z,w\rangle^2}{2\varepsilon^2 \|w\|_2^2}}$ via the Hoeffding's inequality, we obtain

$$\mathbb{E}_{u,w,z,\eta}\left[\|w\|_2^2\,\mathbb{1}_{\langle w,z\rangle\geq 0}\mathbb{1}_{\langle w,z+\eta\rangle\geq 0}\right]=\mathbb{E}_{w,z,\eta}\left[\|w\|_2^2\,\mathbb{1}_{\langle w,z\rangle\geq 0}\mathbb{1}_{\langle w,z+\eta\rangle\geq 0}\right]$$

$$=\mathbb{E}_{w,z}\left[\|w\|_2^2\,\mathbb{1}_{\langle w,z\rangle\geq 0}\mathbb{P}_\eta\left(\langle w,z+\eta\rangle\geq 0\right)\right]$$

$$=\mathbb{E}_{w,z}\left[\|w\|_2^2\,\mathbb{1}_{\langle w,z\rangle\geq 0}\mathbb{P}_\eta\left(\langle w,\eta\rangle\geq -\langle w,z\rangle\right)\right]$$

$$=\mathbb{E}_{w,z}\left[\|w\|_2^2\,\mathbb{1}_{\langle w,z\rangle\geq 0}\mathbb{P}_\eta\left(\langle w,\eta\rangle\leq \langle w,z\rangle\right)\right]$$

$$=\mathbb{E}_{w,z}\left[\|w\|_2^2\,\mathbb{1}_{\langle w,z\rangle\geq 0}\left(1-\mathbb{P}_\eta\left(\langle w,\eta\rangle\geq \langle w,z\rangle\right)\right)\right]$$

$$\geq\mathbb{E}_{w,z}\left[\|w\|_2^2\,\mathbb{1}_{\langle w,z\rangle\geq 0}\left(1-e^{-\frac{\langle w,z\rangle^2}{2\varepsilon^2\|w\|_2^2}}\right)\right]$$

$$=\mathbb{E}_{w,z}\left[\|w\|_2^2\,\mathbb{1}_{\langle w,z\rangle\geq 0}\right]-\mathbb{E}_{w,z}\left[\|w\|_2^2\,\mathbb{1}_{\langle w,z\rangle\geq 0}\,e^{-\frac{\langle w,z\rangle^2}{2\varepsilon^2\|w\|_2^2}}\right]$$

$$=0.5p\sigma_w^2-0.5\,\mathbb{E}_{w,z}\left[\|w\|_2^2\,e^{-\frac{\langle w,z\rangle^2}{2\varepsilon^2\|w\|_2^2}}\right]$$

$$\geq 0.5p\sigma_w^2-0.5\,\mathbb{E}_w\left[\|w\|_2^4\right]^{1/2}\mathbb{E}_{w,z}\left[e^{-\frac{\langle w,z\rangle^2}{\varepsilon^2\|w\|_2^2}}\right]^{1/2}$$

$$=0.5p\sigma_w^2-0.5\sigma_w^2\sqrt{p^2+2p}\,\mathbb{E}_{w,z}\left[e^{-\frac{\langle w,z\rangle^2}{\varepsilon^2\|w\|_2^2}}\right]^{1/2},$$

where the last inequality is obtained via the Cauchy-Schwarz inequality. On the other hand, we have:

$$\mathbb{E}_{u,w,z,\eta}\left[\|w\|_2^2\,\mathbb{1}_{\langle w,z\rangle\geq 0}\mathbb{1}_{\langle w,z+\eta\rangle\geq 0}\right]=\mathbb{E}_{w,z}\left[\|w\|_2^2\,\mathbb{1}_{\langle w,z\rangle\geq 0}\mathbb{P}_\eta\left(\langle w,\eta\rangle\leq \langle w,z\rangle\right)\right]$$

$$\geq\mathbb{E}_{w,z}\left[\|w\|_2^2\,\mathbb{1}_{\langle w,z\rangle\geq 0}0.5\right]=0.25p\sigma_w^2.$$

Now we combine both lower bounds together to establish a lower bound on Eq. ([9](#)):

$$\frac{\mathbb{E}_{w,z,\eta}\left[\|w\|_2^2\,\mathbb{1}_{\langle w,z\rangle\geq 0}\mathbb{1}_{\langle w,z+\eta\rangle\geq 0}\right]}{\sqrt{\mathbb{E}_{w,z}\left[\|w\|_2^2\,\mathbb{1}_{\langle w,z\rangle\geq 0}\right]}\sqrt{\mathbb{E}_{w,z,\eta}\left[\|w\|_2^2\,\mathbb{1}_{\langle w,z+\eta\rangle\geq 0}\right]}}$$

$$\geq\frac{\max\left\{0.5p\sigma_w^2-0.5\sigma_w^2\sqrt{p^2+2p}\,\mathbb{E}_{w,z}\left[e^{-\frac{\langle w,z\rangle^2}{\varepsilon^2\|w\|_2^2}}\right]^{1/2},0.25p\sigma_w^2\right\}}{0.5p\sigma_w^2}$$

$$=\max\left\{1-\sqrt{1+\frac{2}{p}}\,\mathbb{E}_{w,z}\left[e^{-\frac{\langle w/\|w\|_2,z\rangle^2}{\varepsilon^2}}\right]^{1/2},0.5\right\}$$

$$\geq\max\left\{1-\sqrt{2}\,\mathbb{E}_{w,z}\left[e^{-\frac{1}{\varepsilon^2}\langle w/\|w\|_2,z\rangle^2}\right]^{1/2},0.5\right\}, \tag{10}$$

where in the last step we used that $p\geq 2$. $\qquad\square$

The main purpose of obtaining the lower bound in Lemma [2](#) was to get an expression that can give us an insight into the key quantities which gradient alignment at initialization depends on. Considering the limiting case $k,m\to\infty$ was necessary to obtain a ratio of expectations that allowed us to derive a simpler expression. Finally, we lower bounded the gradient alignment from Eq. ([9](#)) using the Hoeffding's and Cauchy-Schwarz inequalities and used $p\geq 2$ to obtain a dimension-independent constant in front of the expectation in Eq. ([10](#)). Now we would like to provide a better understanding about the key quantities involved in the lemma and to assess the tightness of the derived lower bound. For this purpose, in Fig. [9](#) we plot:

**Figure 9:** Visualization of the key quantities involved in Lemma 2.

- $\cos\left(\nabla_x \ell(x,y), \nabla_x \ell(x+\eta,y)\right)$ for $k=100$ patches and $m=4$ filters (which resembles the setting of the 4-filter CNN on CIFAR-10). We note that it is a random variable since it is a function of random variables $x, \eta, W, U$.

- $\lim_{k,m\to\infty} \cos\left(\nabla_x \ell(x,y), \nabla_x \ell(x+\eta,y)\right)$ evaluated via Eq. (9).

- Our first lower bound $\max\left\{1 - \frac{1}{p\sigma_w^2}\, \mathbb{E}_{w,z}\left[\|w\|_2^2\, e^{-\frac{1}{2\varepsilon^2}\langle w/\|w\|_2, z\rangle^2}\right], 0.5\right\}$ obtained via Hoeffding's inequality.

- Our final lower bound $\max\left\{1 - \sqrt{2}\, \mathbb{E}_{w,z}\left[e^{-\frac{1}{\varepsilon^2}\langle w/\|w\|_2, z\rangle^2}\right]^{1/2}, 0.5\right\}$.

For the last three quantities we approximate the expectations by Monte-Carlo sampling by using 1,000 samples. For all the quantities we use patches of size $p = 3 \times 3 \times 3 = 27$ as in our CIFAR-10 experiments. We plot gradient alignment values for $\varepsilon \in [0, 0.1]$ since we are interested in small $\ell_\infty$-perturbations such as, e.g., $\varepsilon = 8/255 \approx 0.03$ which is a typical value used for CIFAR-10 [23]. First, we can observe that all the four quantities have very high values in $[0.7, 1.0]$ for $\varepsilon \in [0, 0.1]$ which is in contrast to the gradient alignment value of 0.12 that we observe after catastrophic overfitting for $\varepsilon = 10/255 \approx 0.04$. Next, we observe that $\cos\left(\nabla_x \ell(x,y), \nabla_x \ell(x+\eta,y)\right)$ has some noticeable variance for the chosen parameters $k = 100$ patches and $m = 4$ filters. However, this variance is significantly reduced when we increase the parameters $k$ and $m$, especially when considering the limiting case $k, m \to \infty$. Finally, we observe that both lower bounds on $\lim_{k,m\to\infty} \cos\left(\nabla_x \ell(x,y), \nabla_x \ell(x+\eta,y)\right)$ that we derived are empirically tight enough to properly capture the behaviour of gradient alignment for small $\varepsilon$. However, we choose to report the last one in the lemma since it is slightly more concise than the one obtained via Hoeffding's inequality.

## B  Experimental details

We list detailed evaluation and training details below.

**Evaluation.** Throughout the paper, we use PGD-50-10 for evaluation of adversarial accuracy which stands for the PGD attack with 50 iterations and 10 random restarts following [47]. We use the step size $\alpha = \varepsilon/4$. The choice of this attack is motivated by the fact that in both public benchmarks of [23] on MNIST and CIFAR-10, the adversarial accuracy of PGD-100-50 and PGD-20-10 respectively is only 2% away from the best entries.

Although we train our models using half precision [24], we always perform robustness evaluation using single precision since evaluation with half precision can sometimes overestimate the robustness of the model due to limited numerical precision in the calculation of the gradients.

We perform evaluation of standard accuracy using full test sets, but we evaluate adversarial accuracy using 1,000 random points on each dataset.

**Training details for ResNet-18.** We use the implementation code of [47] with the only difference that we do not use image normalization and gradient clipping on CIFAR-10 and SVHN since we found that they have no significant influence on the final results. We use cyclic learning rates and

half-precision training following [47]. We do not use random initialization for PGD during adversarial training as we did not find that it leads to any improvements on the considered datasets (see the justifications in Sec. D.1 below). We perform early stopping based on the PGD accuracy on the training set following [47]. We observed that such a simple model selection scheme can successfully select a model before catastrophic overfitting that has non-trivial robustness.

On CIFAR-10, we train all the models for 30 epochs with the maximum learning rate 0.3 except *AT for free* [34] which we train for 96 epochs with the maximum learning rate 0.04 using $m = 8$ minibatch replays to get comparable results to the other methods.

On SVHN, we train all the models for 15 epochs with the maximum learning rate 0.05 except *AT for free* [34] which we train for 45 epochs with the maximum learning rate 0.01 using $m = 8$ minibatch replays. Moreover, in order to prevent convergence to a constant classifier on SVHN, we linearly increase the perturbation radius from 0 to $\varepsilon$ during the first 5 epochs for all methods.

For PGD-2 AT we use for training a 2-step PGD attack with step size $\alpha = \varepsilon/2$, and for PGD-10 AT we use for training a 10-step PGD attack with $\alpha = 2\varepsilon/10$.

For Fig. 1 and Fig. 8 we used the `GradAlign` $\lambda$ values obtained via a linear interpolation on the logarithmic scale between the best $\lambda$ values that we found for $\varepsilon = 8$ and $\varepsilon = 16$ on the test sets. We perform the interpolation on the logarithmic scale since the values of $\lambda$ are non-negative, a usual linear interpolation would lead to negative values of $\lambda$. The resulting $\lambda$ values for $\varepsilon \in \{1, \ldots, 16\}$ are given in Table 2. We note that at the end we do not report the results with $\varepsilon > 12$ for SVHN since many models have trivial robustness close to that of a constant classifier. For the PGD-2 + `GradAlign`

**Table 2:** `GradAlign` $\lambda$ values used for the experiments on CIFAR-10 and SVHN. These values are obtained via a linear interpolation on the logarithmic scale between successful $\lambda$ values at $\varepsilon = 8$ and $\varepsilon = 16$.

| $\varepsilon$ (/255) | 1 | 2 | 3 | 4 | 5 | 6 | 7 | 8 | 9 | 10 | 11 | 12 | 13 | 14 | 15 | 16 |
|---|---|---|---|---|---|---|---|---|---|---|---|---|---|---|---|---|
| $\lambda_{CIFAR-10}$ | 0.03 | 0.04 | 0.05 | 0.06 | 0.08 | 0.11 | 0.15 | 0.20 | 0.27 | 0.36 | 0.47 | 0.63 | 0.84 | 1.12 | 1.50 | 2.00 |
| $\lambda_{SVHN}$ | 1.66 | 1.76 | 1.86 | 1.98 | 2.10 | 2.22 | 2.36 | 2.50 | 2.65 | 2.81 | 2.98 | 3.16 | 3.35 | 3.56 | 3.77 | 4.00 |

experiments reported below in Table 4 and Table 5, we use $\lambda = 0.1$ for the CIFAR-10 and $\lambda = 0.5$ for SVHN experiments.

**Training details for the single-layer CNN.** The single-layer CNN that we study in Sec. 4 has 4 convolutional filters, each of them of size $3 \times 3$. After the convolution we apply ReLU activation, and then we directly have a fully-connected layer, i.e. we do not use any pooling layer. For training we use the ADAM optimizer with learning rate 0.003 for 30 epochs using the same cyclical learning rate schedule.

**ImageNet experiments.** We use ResNet-50 following the training scheme of [47] which includes 3 training stages on different image resolution. For `GradAlign`, we slightly reduce the batch size on the second and third stages from 224 and 128 to 180 and 100 respectively in order to reduce the memory consumption. For all $\varepsilon \in \{2, 4, 6\}$, we train FGSM models with `GradAlign` using $\lambda \in \{0.01, 0.1\}$. The final $\lambda$ we report are $\lambda \in \{0.01, 0.01, 0.1\}$ for $\varepsilon \in \{2, 4, 6\}$ respectively.

**Computing infrastructure.** We perform all our experiments on NVIDIA V100 GPUs with 32GB of memory.

# C  Supporting experiments and visualizations for Sec. 3 and Sec. 4

We describe here supporting experiments and visualizations related to Sec. 3 and Sec. 4.

## C.1  Quality of the linear approximation for ReLU networks

For the loss function $\ell$ of a ReLU-network, we compute empirically the quality of the linear approximation defined as

$$| \ell(x + \delta) - \ell(x) - \langle \delta, \nabla_x \ell(x) \rangle |,$$

where the dependency of the loss $\ell$ on the label $y$ and parameters $\theta$ are omitted for clarity. Then we perform the following experiment: we take a perturbation $\delta \in \{-\varepsilon, \varepsilon\}^d$, and then zero out different

**a.) Standard model**

**b.) PGD-trained model**

**Figure 10:** The quality of the linear approximation of $\ell(x + \delta)$ for $\delta$ with different $\ell_2$-norm for $\|\delta\|_\infty$ fixed to $\varepsilon$ for a standard and PGD-trained ResNet-18 on CIFAR-10.

fractions of its coordinates, which leads to perturbations with a fixed $\|\delta\|_\infty = \varepsilon$, but with different $\|\delta\|_2 \in [0, \sqrt{d}\varepsilon]$. As the starting $\delta$ we choose two types of perturbations: $\delta_{FGSM}$ generated by FGSM and $\delta_{random}$ sampled uniformly from the corners of the $\ell_\infty$-ball. We plot the results in Fig. 10 on CIFAR-10 for $\varepsilon = 8/255$ averaged over 512 test points, and conclude that for both $\delta_{FGSM}$ and $\delta_{random}$ the validity of the linear approximation crucially depends on $\|\delta\|_2$ even when $\|\delta\|_\infty$ is fixed. The phenomenon is even more pronounced for FGSM perturbations as the linearization error is much higher there. Moreover, this observation is consistent across both standardly and adversarially trained ResNet-18 models.

### C.2   Catastrophic overfitting in a single-layer CNN

We describe here complementary figures to Sec. 4 which are related to the single-layer CNN.

**Training curves.** In Fig. 11, we show the evolution of the FGSM/PGD accuracy, FGSM/PGD loss, and gradient alignment together with $\cos(\delta_{FGSM}, \delta_{PGD})$. We observe that catastrophic overfitting occurs around epoch 6 and that its pattern is the same as for the deep ResNet which was illustrated in Fig. 4. Namely, we see that concurrently the following changes occur around epoch 6: (a) there is a sudden drop of PGD accuracy with an increase in FGSM accuracy, (b) the PGD loss grows by an order of magnitude while the FGSM loss decreases, (c) both gradient alignment and $\cos(\delta_{FGSM}, \delta_{PGD})$ significantly decrease. Throughout all our experiments we observe a very high correlation between $\cos(\delta_{FGSM}, \delta_{PGD})$ and gradient alignment. This motivates our proposed regularizer `GradAlign` which relies on the cosine between $\nabla_x \ell(x, y; \theta)$ and $\nabla_x \ell(x + \eta, y; \theta)$, where $\eta$ is a *random* point. In this way, we avoid using an iterative procedure inside the regularizer unlike, for example, the approach of [28].

**Additional filters.** In Fig. 12, we show the evolution of the regular filter $w_1$ and filter $w_4$ that leads to catastrophic overfitting for the three input channels (red, green, blue). We can observe that in the red and green channels, $w_4$ has learned a Laplace filter which is very sensitive to noise. Moreover, $w_4$ significantly increases in magnitude after catastrophic overfitting contrary to $w_1$ whose magnitude only decreases (see the colorbar values in Fig. 12 and the plots in Fig. 5).

**Additional feature maps.** In Fig. 13, we show additional feature maps for images with and without uniform random noise $\eta \sim \mathcal{U}([-10/255, 10/255]^d)$. These figures complement Fig. 7 shown in the main part. We clearly see that only the last filter $w_4$ is sensitive to the noise since the feature maps change dramatically. At the same time, other filters $w_1$, $w_2$, $w_3$ are only slightly affected by the

**Figure 11:** Visualization of the training process of an FGSM trained CNN with 4 filters with $\varepsilon = 10/255$. We can observe catastrophic overfitting around epoch 6.

**Figure 12:** Evolution of the regular filter $w_1$ and filter $w_4$ that leads to catastrophic overfitting. We plot red (R), green (G), and blue (B) channels of the filters. We can observe that in R and G channels, $w_4$ has learned a Laplace filter which is very sensitive to noise.

addition of the noise. We also show the input gradients in the last column which illustrate that after adding the noise the gradients change drammatically which leads to small gradient alignment and, in turn, to the failure of FGSM as the solution of the inner maximization problem.

**Figure 13:** Input images, feature maps, and gradients of the single-layer CNN trained on CIFAR-10 at the end of training (after catastrophic overfitting). *Odd row*: original images. *Even row*: original image plus random noise $\mathcal{U}([-^{10}/_{255}, {}^{10}/_{255}]^d)$. We observe that only the last filter $w_4$ is highly sensitive to the small uniform noise since the feature maps change dramatically.

# D  Additional experiments for different adversarial training schemes

In this section, we describe additional experiments related to `GradAlign` that complement the results shown in Sec. 5.

## D.1  Stronger PGD-2 baseline

As mentioned in Sec. 5, the PGD-2 training baseline that we report outperforms other similar baselines reported in the literature [49, 28]. Here we elaborate what are likely to be the most important sources of difference. First, we follow the cyclical learning rate schedule of [47] which can work as implicit early stopping and thus can help to prevent catastrophic overfitting observed for PGD-2 in [28]. Another source of difference is that [28] use the ADAM optimizer while we stick to the standard PGD updates using the sign of the gradient [23].

The second important factor is a proper step size selection. While [49] do not observe catastrophic overfitting, their PGD-3 baseline achieves only $32.51\%$ adversarial accuracy compared to the $48.43\%$ for our PGD-2 baseline evaluated with a stronger attack (PGD-50-10 instead of PGD-20-1). One potential explanation for this difference lies in the step size selection, where for PGD-2 we use $\alpha = \varepsilon/2$. Related to the step size selection, we also found that using random initialization in PGD (we will refer to as PGD-k-RS) as suggested in [23] requires a larger step size $\alpha$. We show the results in Table 3 where we can see that PGD-2-RS AT with $\alpha = \varepsilon/2$ achieves suboptimal robustness compared to $\alpha = \varepsilon$ used for training. However, we consistently observed that PGD-2 AT with $\alpha = \varepsilon/2$ and *no random step* performs best. Thus, we use the latter as our PGD-2 baseline throughout the paper, thus always starting PGD-2 from the original point, without using any random step.

**Table 3:** Robustness of different PGD-2 schemes for $\varepsilon = 8/255$ on CIFAR-10 for ResNet-18. The results are averaged over 5 random seeds used for training.

| Model | PGD-2-RS AT, $\alpha = \varepsilon/2$ | PGD-2-RS AT, $\alpha = \varepsilon$ | PGD-2 AT, $\alpha = \varepsilon/2$ |
|---|---|---|---|
| **PGD-50-10 accuracy** | 45.06±0.44% | 48.07±0.52% | 48.43±0.40% |

## D.2  Results with early stopping

We complement the results presented in Fig. 8 *without early stopping* with the results *with early stopping* which we show in Fig. 14. For CIFAR-10, we observe that FGSM + `GradAlign` leads to a good robustness and accuracy outperforming FGSM AT and FGSM-RS AT and performing similarly to PGD-2 and slightly improving for larger $\varepsilon$ close to $16/255$. For SVHN, `GradAlign` leads to better robustness than other FGSM-based methods. We also observe that for large $\varepsilon$ on both CIFAR-10 and SVHN, *AT for Free* performs similarly to FGSM-based methods. Moreover, for $\varepsilon \geq 10/255$ on SVHN, *AT for Free* converges to a constant classifier.

**Figure 14:** Accuracy (dashed line) and robustness (solid line) of different adversarial training (AT) methods on CIFAR-10 and SVHN with ResNet-18 trained and evaluated with different $l_\infty$-radii. The results are obtained **with early stopping**, averaged over 5 random seeds used for training and reported with the standard deviation.

On both CIFAR-10 and SVHN, we can see that although early stopping can lead to non-trivial robustness, standard accuracy is often significantly sacrificed which limits the usefulness of this technique. This is in contrast to training with `GradAlign` which leads to the same standard accuracy as PGD-10 training.

### D.3 Results for specific $\ell_\infty$-radii

Here we report results from Fig. 8 for specific $\ell_\infty$-radii which are most often studied in the literature.

**CIFAR-10 results.** We report robustness and accuracy in Table 4 for CIFAR-10 without using early stopping where we can clearly see which methods lead to catastrophic overfitting and thus suboptimal robustness. We compare the same methods as in Fig. 8, and additionally we report the results for $\varepsilon = 8/255$ of the CURE [25], YOPO [49], and LLR [28] approaches. First, for $\varepsilon = 8/255$, we see that FGSM + `GradAlign` outperforms *AT for Free* and all methods that use FGSM training. Then, we also observe that the model trained with CURE [25] leads to robustness that is suboptimal compared to FGSM-RS AT evaluated with a stronger attack: $36.3\%$ vs $45.1\%$. YOPO-3-5 and YOPO-5-3 [49] require 3 and 5 full steps of PGD respectively, thus they are much more expensive than FGSM-RS AT, and, however, they lead to worse adversarial accuracy: $38.18\%$ and $44.72\%$ vs $45.10\%$. Qin et al. [28] report that LLR-2, i.e. their approach with 2 steps of PGD, achieves $44.50\%$ adversarial accuracy with MultiTargeted attack [13] and $46.47\%$ with their untargeted PGD attack which uses a different loss function compared to our PGD attack. These two evaluations are not directly comparable to other results in Table 4 since the attacks are different and moreover they use a larger network (Wide-ResNet-28-8) which usually leads to better results [23]. However, we think that the gap of $3 - 4\%$ adversarial accuracy of MultiTargeted evaluation compared to that of our reported FGSM + `GradAlign` and PGD-2 methods ($47.58\%$ and $48.43\%$ resp.) is still significant since the difference between MultiTargeted and a PGD attack with random restarts is observed to be small (e.g. around 1% between MultiTargeted and PGD-20-10 on the CIFAR-10 challenge of [23]).

For $\varepsilon = 16/255$, none of the one-step methods work without early stopping except FGSM + `GradAlign`. We also evaluate PGD-2 + `GradAlign` and conclude that the benefit of combining the two comes when PGD-2 alone leads to catastrophic overfitting which occurs at

**Table 4:** Robustness and accuracy of different robust training methods on **CIFAR-10**. We report results without early stopping for ResNet-18 unless specified otherwise in parentheses. The results of all the methods reported in Fig. 8 are shown here with the standard deviation and averaged over 5 random seeds used for training.

| Model | Accuracy | | Attack |
|---|---|---|---|
| | Standard | Adversarial | |
| $\varepsilon = 8/255$ | | | |
| Standard | 94.03% | 0.00% | PGD-50-10 |
| CURE [25] | 81.20% | 36.30% | PGD-20-1 |
| YOPO-3-5 [49] | 82.14% | 38.18% | PGD-20-1 |
| YOPO-5-3 [49] | 83.99% | 44.72% | PGD-20-1 |
| LLR-2 (Wide-ResNet-28-8) [28] | 90.46% | 44.50% | MultiTargeted [28] |
| FGSM | 85.16±1.3% | 0.02±0.04% | PGD-50-10 |
| FGSM-RS | 84.32±0.08% | 45.10±0.56% | PGD-50-10 |
| FGSM + `GradAlign` | 81.00±0.37% | **47.58±0.24%** | PGD-50-10 |
| AT for Free ($m = 8$) | 77.92±0.65% | 45.90±0.98% | PGD-50-10 |
| PGD-2 ($\alpha = 4/255$) | 82.15±0.48% | 48.43±0.40% | PGD-50-10 |
| PGD-2 ($\alpha = 4/255$) + `GradAlign` | 81.16±0.39% | 47.76±0.77% | PGD-50-10 |
| PGD-10 ($\alpha = 2\varepsilon/10$) | 81.88±0.37% | **50.04±0.79%** | PGD-50-10 |
| $\varepsilon = 16/255$ | | | |
| FGSM | 73.76±7.4% | 0.00±0.00% | PGD-50-10 |
| FGSM-RS | 72.18±3.7% | 0.00±0.00% | PGD-50-10 |
| FGSM + `GradAlign` | 58.46±0.22% | **28.88±0.70%** | PGD-50-10 |
| AT for Free ($m = 8$) | 48.10±9.83% | 0.00±0.00% | PGD-50-10 |
| PGD-2 ($\alpha = \varepsilon/2$) | 68.65±5.83% | 9.92±14.00% | PGD-50-10 |
| PGD-2 ($\alpha = \varepsilon/2$) + `GradAlign` | 61.38±0.71% | 29.80±0.42% | PGD-50-10 |
| PGD-10 ($\alpha = 2\varepsilon/10$) | 60.28±0.50% | **33.24±0.52%** | PGD-50-10 |

**Table 5:** Robustness and accuracy of different robust training methods on **SVHN**. We report results without early stopping for ResNet-18. All the results are reported with the standard deviation and averaged over 5 random seeds used for training.

| Model | Accuracy | |
|---|---|---|
| | Standard | PGD-50-10 |
| $\varepsilon = 8/255$ | | |
| Standard | 96.00% | 1.00% |
| FGSM | 91.40±1.64% | 0.04±0.05% |
| FGSM-RS | 95.38±0.27% | 0.00±0.00% |
| FGSM + GradAlign | 92.36±0.47% | **42.08±0.25%** |
| AT for Free ($m = 8$) | 75.34±28.4% | 43.16±12.3% |
| PGD-2 ($\alpha = \varepsilon/2$) | 92.68±0.45% | 47.28±0.26% |
| PGD-2 + GradAlign ($\alpha = \varepsilon/2$) | 92.46±0.35% | 47.02±0.83% |
| PGD-10 ($\alpha = 2\varepsilon/10$) | 91.92±0.40% | **52.08±0.49%** |
| $\varepsilon = 12/255$ | | |
| FGSM | 88.74±1.25% | 0.00±0.00% |
| FGSM-RS | 94.70±0.66% | 0.00±0.00% |
| FGSM + GradAlign | 88.54±0.21% | **24.04±0.31%** |
| AT for Free ($m = 8$) | 18.50±0.00% | 18.50±0.00% |
| PGD-2 ($\alpha = \varepsilon/2$) | 92.74±2.26% | 14.30±13.34% |
| PGD-2 + GradAlign ($\alpha = \varepsilon/2$) | 87.14±0.26% | 31.26±0.24% |
| PGD-10 ($\alpha = 2\varepsilon/10$) | 84.52±0.63% | **38.32±0.38%** |

**Table 6:** Robustness and accuracy of different robust training methods on **ImageNet**. We report results without early stopping for ResNet-50.

| Model | $\ell_\infty$-radius | Standard accuracy | PGD-50-10 accuracy |
|---|---|---|---|
| FGSM | 2/255 | 61.7% | 42.1% |
| FGSM-RS | 2/255 | 59.3% | 41.1% |
| FGSM + GradAlign | 2/255 | 61.8% | 41.4% |
| FGSM | 4/255 | 56.9% | 30.6% |
| FGSM-RS | 4/255 | 55.3% | 27.8% |
| FGSM + GradAlign | 4/255 | 57.8% | 30.5% |
| FGSM | 6/255 | 51.5% | 20.6% |
| FGSM-RS | 6/255 | 36.6% | 0.1% |
| FGSM + GradAlign | 6/255 | 51.5% | 20.3% |

$\varepsilon = 16/255$. For $\varepsilon = 8/255$, there is no benefit of combining the two approaches. This is consistent with our observation regarding catastrophic overfitting for FGSM (e.g. see Fig. 8 for small $\varepsilon$): if there is no catastrophic overfitting, there is no benefit of adding GradAlign to FGSM training.

To further ensure that FGSM + GradAlign models do not benefit from gradient masking [27], we additionally compare the robustness of FGSM + GradAlign and FGSM-RS models obtained via *AutoAttack* [8]. We observe that *AutoAttack* proportionally reduces the adversarial accuracy of both models: for $\varepsilon = 8/255$, FGSM + GradAlign achieves 44.54±0.24% adversarial accuracy while FGSM-RS achieves 42.80±0.58%. This is consistent with the evaluation results of [8] where they show that *AutoAttack* reduces adversarial accuracy for many models by 2%-3% for $\varepsilon = 8/255$ compared to the originally reported results based on the standard PGD attack (see Table 2 in [8]). The same tendency is observed also for higher $\varepsilon$, e.g. for $\varepsilon = 16/255$ FGSM + GradAlign achieves 20.56±0.36% adversarial accuracy when evaluated with *AutoAttack*.

**SVHN results.** We report robustness and accuracy in Table 5 for SVHN without using early stopping. We can see that for both $\varepsilon = 8/255$ and $\varepsilon = 16/255$, GradAlign successfully prevents catastrophic overfitting in contrast to FGSM and FGSM-RS, although there is still a 5% gap to PGD-2 training for $\varepsilon = 8/255$. *AT for free* performs slightly better than FGSM + GradAlign for $\varepsilon = 8/255$, but it already starts to show a high variance in the robustness and accuracy depending on the random seed. For $\varepsilon = 12/255$, all the 5 models of *AT for free* converge to a constant classifier.

Combining PGD-2 with `GradAlign` does not lead to improved results for $\varepsilon = {}^8/255$ since there is no catastrophic overfitting for PGD-2. However, for $\varepsilon = {}^{12}/255$, we can clearly see that PGD-2 + `GradAlign` leads to better results than PGD-2 achieving $31.26\pm0.24\%$ instead of $14.30\pm13.34\%$ adversarial accuracy.

**ImageNet results.** We also perform similar experiments on ImageNet in Table 6. We observe that even for standard FGSM training, catastrophic overfitting *does not* occur for $\varepsilon \in \{{}^2/255, {}^4/255\}$ considered in [34, 47], and thus there is no additional benefit from using `GradAlign` since its main role is to prevent catastrophic overfitting. We report the results of FGSM + `GradAlign` for completeness to show that `GradAlign` can be applied on the ImageNet scale, although it leads to approximately $3\times$ slowdown on ImageNet compared to standard FGSM training.

For $\varepsilon = {}^6/255$, we observe that catastrophic overfitting occurs for FGSM-RS very early in training (around epoch 3), but not for FGSM or FGSM + `GradAlign` training. This contradicts our observations on CIFAR-10 and SVHN where we observed that FGSM-RS usually helps to postpone catastrophic overfitting to higher $\varepsilon$. However, it is computationally demanding to replicate the results on ImageNet multiple times over different random seeds as we did for CIFAR-10 and SVHN. Thus, we leave a more detailed investigation of catastrophic overfitting on ImageNet for future work.

### D.4 Ablation studies

In this section, we aim to provide more details about sensitivity of `GradAlign` to its hyperparameter $\lambda$, the total number of training epochs, and also discuss training with `GradAlign` for very high $\varepsilon$ values.

**Ablation study for GradAlign $\lambda$.** We provide an ablation study for the regularization parameter $\lambda$ of `GradAlign` in Fig. 15, where we plot the adversarial accuracy of ResNet-18 trained using FGSM + `GradAlign` with $\varepsilon = {}^{16}/255$ on CIFAR-10. First, we observe that for small $\lambda$ catastrophic overfitting occurs so that the average PGD-50-10 accuracy is either $0\%$ or greater than $0\%$ but has a high standard deviation since only some runs are successful while other runs fail because of catastrophic overfitting. We observe that the best performance is achieved for $\lambda = 2$ where catastrophic overfitting does not occur and the final adversarial accuracy is very concentrated. For larger $\lambda$ values we observe a slow decrease in the adversarial accuracy since the model becomes overregularized. We note that the range of $\lambda$ values which have close to the best performance ($\geq 26\%$ adversarial accuracy) ranges in $[0.25, 4]$, thus we conclude that `GradAlign` is robust to the exact choice of $\lambda$. This is also confirmed by our hyperparameter selection method for Fig. 8, where we performed a linear interpolation on the logarithmic scale between successful $\lambda$ values for $\varepsilon = {}^8/255$ and $\varepsilon = {}^{16}/255$. Even such a coarse hyperparameter selection method, could ensure that none of the FGSM + `GradAlign` runs reported in Fig. 15 suffered from catastrophic overfitting.

**Figure 15:** Ablation study for the regularization parameter $\lambda$ for FGSM + `GradAlign` under $\varepsilon = {}^{16}/255$ without early stopping. We train ResNet-18 models on CIFAR-10. The results are averaged over 3 random seeds used for training and reported with the standard deviation.

**Figure 16:** Ablation study for the total number of training epochs for FGSM + `GradAlign` under $\varepsilon = {}^8/255$ without early stopping. We train ResNet-18 models on CIFAR-10. The results are averaged over 3 random seeds used for training and reported with the standard deviation.

**Ablation study for the total number of training epochs.** Recently, Rice et al. [30] brought up the importance of early stopping in adversarial training. They identify the phenomenon called *robust overfitting* when training longer hurts the adversarial accuracy on the test set. Thus, we check here whether training with `GradAlign` has some influence on robust overfitting. We note that the authors of [30] suggest that robust and catastrophic overfitting phenomena are distinct since robust overfitting implies a gap between training and test set robustness, while catastrophic overfitting implies low robustness on *both* training and test sets. To explore this for FGSM + `GradAlign`, in Fig. 16 we show the final clean and adversarial accuracies for five different models trained with $\{30, 50, 100, 150, 250\}$ epochs. We observe the same trend as [30] report: training longer slightly degrades adversarial accuracy (while in our case also the clean accuracy slightly improves). Thus, this experiment also suggests that robust overfitting is not directly connected to catastrophic overfitting and has to be addressed separately. Finally, we note based on Fig. 16 that when we use FGSM in combination with `GradAlign`, even training *up to 200 epochs* does not lead to catastrophic overfitting.

**Ablation study for very high $\varepsilon$.** Here we make an additional test on whether `GradAlign` prevents catastrophic overfitting for very high $\varepsilon$ values. In Fig. 8 and Fig. 14 we showed results for $\varepsilon \leq 16$ for CIFAR-10 and for $\varepsilon \leq 12$ on SVHN. For SVHN, FGSM + `GradAlign` achieves 24.04$\pm$0.31% adversarial accuracy which is already close to that of a majority classifier (18.50%). The effect of increasing the perturbations size $\varepsilon$ on SVHN even further just leads to learning a constant classifier. However, on CIFAR-10 for $\varepsilon = 16$, FGSM + `GradAlign` achieves 28.88$\pm$0.70% adversarial accuracy which is sufficiently far from that of a majority classifier (10.00%). Thus, a natural question is whether catastrophic overfitting still occurs for `GradAlign` on CIFAR-10, but just for higher $\varepsilon$ values than what we considered in the main part of the paper. To show that it is not the case, in Table 7 we show the results of FGSM + `GradAlign` trained with $\varepsilon \in \{^{24}/_{255}, ^{32}/_{255}\}$ (we use $\lambda = 2.0$ and the maximum learning rate 0.1). We observe no signs of catastrophic overfitting *even for very high $\varepsilon$* such as $^{32}/_{255}$. Note that in this case the standard accuracy is very low (23.07$\pm$3.35%), thus considering such large perturbations is not practically interesting, but it rather serves as a sanity check that our method does not suffer from catastrophic overfitting even for very high $\varepsilon$.

**Table 7:** Robustness and accuracy of FGSM + `GradAlign` for very high $\varepsilon$ on CIFAR-10 without early stopping for ResNet-18. We report results with the standard deviation and averaged over 3 random seeds used for training. We observe no catastrophic overfitting even for very high $\varepsilon$.

| $\ell_\infty$-radius | Standard accuracy | PGD-50-10 accuracy |
|---|---|---|
| 24/255 | 41.80$\pm$0.36% | 17.07$\pm$0.90% |
| 32/255 | 23.07$\pm$3.35% | 12.93$\pm$1.44% |

### D.5  Comparison of GradAlign to gradient-based penalties

In this section, we compare `GradAlign` to other alternatives: $\ell_2$ gradient norm penalization and CURE [25]. The motivation to study them comes from the fact that after catastrophic overfitting, the input gradients change dramatically inside the $\ell_\infty$-balls around input points, and thus other gradient-based regularizers may also be able to improve the stability of the input gradients and thus prevent catastrophic overfitting.

In Table 8, we present results of FGSM training with other gradient-based penalties studied in the literature:

- $\ell_2$ gradient norm regularization [31, 36]: $\lambda \left\| \nabla_x \ell(x, y; \theta) \right\|_2^2$,

- curvature regularization (CURE) [25]: $\lambda \left\| \nabla_x \ell(x + \delta_{FGSM}, y; \theta) - \nabla_x \ell(x, y; \theta) \right\|_2^2$.

First of all, we note that the originally proposed approaches [31, 36, 25] *do not* involve adversarial training and rely *only* on these gradient penalties to achieve some degree of robustness. In contrast, we *combine* the gradient penalties with FGSM training to see whether they can prevent catastrophic overfitting similarly to `GradAlign`. For the gradient norm penalty, we use the regularization parameters $\lambda \in \{1,000, 2,000\}$ for $\varepsilon \in \{^8/_{255}, ^{16}/_{255}\}$ respectively. For CURE, we use $\lambda \in \{700, 20,000\}$ for $\varepsilon \in \{^8/_{255}, ^{16}/_{255}\}$ respectively. In both cases, we found the optimal hyperparameters using a

**Table 8:** Additional comparison of FGSM AT with `GradAlign` to FGSM AT with other gradient penalties on CIFAR-10. We report results without early stopping for ResNet-18. All the results are reported with the standard deviation and averaged over 5 random seeds used for training.

| Model | Accuracy | |
|---|---|---|
| | Standard | PGD-50-10 |
| $\varepsilon = 8/255$ | | |
| FGSM + $\|\nabla_x\|_2^2$ | 77.47±0.14% | 46.69±1.27% |
| FGSM + CURE | 80.20±0.29% | 47.25±0.21% |
| FGSM + `GradAlign` | 81.00±0.37% | **47.58±0.24%** |
| $\varepsilon = 16/255$ | | |
| FGSM + $\|\nabla_x\|_2^2$ | 56.44±2.22% | 13.64±11.2% |
| FGSM + CURE | 62.39±0.42% | 25.38±0.29% |
| FGSM + `GradAlign` | 58.46±0.22% | **28.88±0.70%** |

grid search over $\lambda$. We can see that for $\varepsilon = 8/255$ all three approaches successfully prevent catastrophic overfitting, although the final robustness slightly varies between 46.69% for FGSM with the $\ell_2$-gradient penalty and 47.58% for FGSM with `GradAlign`.

For $\varepsilon = 16/255$, both FGSM + CURE and FGSM + `GradAlign` prevent catastrophic overfitting leading to very concentrated results with a small standard deviation (0.29% and 0.70% respectively). However, the average adversarial accuracy is better for FGSM + `GradAlign`: 28.88% vs 25.38%. At the same time, FGSM with the $\ell_2$-gradient penalty leads to unstable final performance: the adversarial accuracy has a high standard deviation: $13.64 \pm 11.2\%$.

We think that the main difference in the performance of `GradAlign` compared to the gradient penalties that we considered comes from the fact that it is invariant to the gradient norm, and it takes into account only the directions of two gradients inside the $\ell_\infty$-ball around the given input.

Inspired by CURE, we also tried two additional experiments:

1. Using the FGSM point $\delta_{FGSM}$ for the gradient taken at the second input point for `GradAlign`, but we observed that it does not make a substantial difference, i.e. this version of `GradAlign` also prevents catastrophic overfitting and leads to similar results. However, if we use CURE without FGSM in the cross-entropy loss, then we observe a benefit of using $\delta_{FGSM}$ in the regularizer which is consistent with the observations made in Moosavi-Dezfooli et al. [25].

2. Using `GradAlign` without FGSM in the cross-entropy loss. In this case, we observed that the model did not significantly improve its robustness suggesting that `GradAlign` *is not* a sufficient regularizer on its own to promote robustness and has to be used *with* some adversarial training method.

We think that an interesting future direction is to explore how one can speed up `GradAlign` or to come up with other regularization methods that are also able to prevent catastrophic overfitting, but avoid relying on the input gradients which lead to a slowdown in training. We think that some potential strategies to speed up `GradAlign` can include parallelization of the computations or saving some computations by subsampling the training batches for the regularizer. We postpone a further exploration of these ideas to future work.