[Reviews · NeurIPS 2020]

Review 1

Summary and Contributions: The paper suggest a regularization method called GradAlign that prevents catastrophic overfitting of single gradient based adversarial training.

Strengths: The proposed method successfully prevents catastrophic overfitting. Discussion on a single CNN helps understand the concept of the gradient alignment. Extensive experiments on previous work are also the strengths of the paper.

Weaknesses: The proposed method takes twice longer than single gradient based training methods as mentioned by authors. Exact runtime comparison would improve the completeness of the paper. Moreover, there should be more discussion on a detrimental noise-sensitive filter learned during FGSM training. Why a noise-sensitive filter is learned by FGSM training? ------------------------------------------------ After rebuttal I would like to keep my score. Though, I have still some concerns after rebuttal. They said "GradAlign leads 2x slowdown compared to FGSM-AT", The author should changed this to "3x" as author mentioned in rebuttal. I think it is large gap between 2x and 3x in this area of research (Fast training). If it is 3x slower than FGSM-AT, it should be compared to PGD-3 AT. However, in overall, the paper address the problem of catastrophic overfitting with the appropriate analysis. I agree other reviewers that additional experiments with large epsilon on ImageNet (ex, eps=8,16 as Xie et al) would improve the quality of the paper. Xie, Cihang, et al. "Feature denoising for improving adversarial robustness." Proceedings of the IEEE Conference on Computer Vision and Pattern Recognition. 2019.

Correctness: The claims appear to be correct based on the results. The empirical methodology is correct.

Clarity: The paper is well written.

Relation to Prior Work: Prior work (Wong et al. ICLR 2020) uses early stopping to avoid catastrophic overfitting while the paper suggests a regularization without early stopping.

Reproducibility: Yes

Additional Feedback: Exact runtime comparison would improve the completeness of the paper. What makes the gradient alginment low by FGSM training? Why a noise-sensitive filter is learned by FGSM training?


Review 2

Summary and Contributions: Update (post-rebuttal): Thank you for the rebuttal. After careful reading and considering the opinions of fellow reviewers, I do not see any reason to update my score. An understanding of the limits of the authors' approach would be helpful. In particular, whether the technique works for high-dimensional inputs. ====== In this paper, the authors study the phenomenon of catastrophic overfitting (happening when FGSM is used for adversarial training). They discover that the main culprit is the lack of gradient alignment and propose a novel regularizer (named GradAlign). This regularizer can be used in conjunction with FGSM to train robust models without encountering catastrophic overfitting. The experimental results support the authors claims.

Strengths: (a) The paper is well written and provides ample evidence of the soundness of the authors' approach. (b) Understanding how and when catastrophic overfitting happens during adversarial training is important, and any approach that makes the process of training robust models faster is relevant to NeurIPS community. (c) The idea in itself (that aligning gradients is important to avoid gradient obfuscation) is not particularly novel ([26] explains it quite well), but the execution of the idea is innovative and the analysis is thorough. [26] C. Qin et al., “Adversarial Robustness through Local Linearization”

Weaknesses: (a) Apart from the analysis, a key point that the paper makes is that runtime is not significantly impacted by the new regularizer. If so, it would be helpful to provide runtime performance (even if only present in the Appendix). I gather that GradAlign+FGSM is about 3x slower than FGSM (due to GradAlign having to take a double-gradient). Is this correct? (b) The experimental results could benefit from running stronger attacks (that are less affected by mild gradient obfuscation). E.g., AutoAttack [1] or MultiTargeted [2]. I believe this is not crucial as it's unlikely that the gradients are obfuscated by GradAlign, but would be good to have. Adding a loss landscape visualization (e.g., Fig1 in [26]) could be helpful. (c) I find the citations of [26] (known as LLR) and [24] (known as CURE) to come rather late in the manuscript. Their exact goal is to make the loss surface smoother which also the goal of this paper. In fact, both of these papers do not rely on adversarial training and are quite competitive with FGSM+GradAlign. For example, LLR-2 achieves 46.47% against a standard untargeted attack (possibly PGD-100 with 1 restarts) while FGSM+GradAlign achieves 47.58% (against PGD-50-10). [26] C. Qin et al., “Adversarial Robustness through Local Linearization” [1] F. Croce and M. Hein, “Reliable evaluation of adversarial robustness with an ensemble of diverse parameter-free attacks” [24] S.-M. Moosavi-Dezfooli, A. Fawzi, J. Uesato, and P. Frossard, “Robustness via curvature regularization, and vice versa” [2] S. Gowal, J. Uesato, C. Qin, P.-S. Huang, T. Mann, and P. Kohli, “An Alternative Surrogate Loss for PGD-based Adversarial Testing”

Correctness: The claims are correct and the empirical methodology is correct. See above section on how to strengthen the results.

Clarity: The paper is very clear and well written.

Relation to Prior Work: The prior work is adequately discussed. However, I would introduce [24] and [26] in the related work section rather in the middle of the explanatory text.

Reproducibility: Yes

Additional Feedback: First, I'd like to start by saying that I enjoyed reading the paper. In particular, the last 2 paragraphs of section 3 are well delivered. The notion that only gradient alignment (rather than linearity) is sufficient makes a lot of sense a posteriori. Here are few additional points: (a) In Eq. (3), what does the product sign mean? (b) Line 118: How is the alpha step-size tuned for this experiments? (is it 1.25*\epsilon?) (c) Line 192: This statement about LLR is somewhat overly negative. The LLR paper clearly demonstrates that less iterations are needed to obtain comparable accuracy to PGD AT; and since the authors method seems on par in terms of accuracy to LLR-2, it seems to be an unfair point. (d) An open question is how well FGSM+GradAlign would perform in large epsilon settings (16/255) on ImageNet against random targeted attacks as done in [26]. It might be out of scope of this paper, but I'm genuinely interested in understanding whether the random sampling of \eta could be problematic for high-dimensional inputs. The principle behind LLR is that it will try to find a worse-case violation of the linearity measure. (e) In the Appendix, LLR is listed with the MultiTargeted attack which has been shown to be significantly better than standard PGD (especially when gradients are locally aligned) [2]. Maybe it would be good to list it with the untargeted attack which acheives 46.47%. If the authors intend to cite MultiTargeted, it would probably best to cite [2] rather than [26]. [2] S. Gowal, J. Uesato, C. Qin, P.-S. Huang, T. Mann, and P. Kohli, “An Alternative Surrogate Loss for PGD-based Adversarial Testing”


Review 3

Summary and Contributions: This paper proposed an improved fast adversarial training method for robust models. The authors first show that when perturbation \epsilon is large, previous fast adversarial training methods rely on early stop for robustness. Training longer leads to almost 0 robust accuracy, which is called "catastrophic overfitting". Catastrophic overfitting correlates with cosine distance of two "nearby" gradients and adding such a regularizer could train robust models for large \epsilon without early stop.

Strengths: I like figure 1 a lot. Figure 1 clearly shows that without early stop, the previous fast adversarial training methods fail, and the proposed method works for large perturbation \epsilon ( > 8 for CIFAR-10). I also like Figure 4 that shows the correlation between sudden drop of robust accuracy (catastrophic overfitting) and the change in the cosine distance that will be used as a regularizer.

Weaknesses: My main concern is the practical contribution of the paper. The regularizer is based on gradient difference, and training with such a regularizer often needs double-backprop (i.e., backprop on gradients), which in my experience is rather slow. The proposed method based on FGSM + regularizer could be even slower than multi-step PGD. In this case, I would vote for early stop since evaluating for early stop is not very complicated and rather efficient.

Correctness: Mostly true. Concerns about efficiency (training time) claims.

Clarity: Yes

Relation to Prior Work: Yes

Reproducibility: Yes

Additional Feedback: I hope the following relatively minor comments would help the authors improve the paper. 1) I have mixed feelings about Section 3 on explaining why previous Fast FGSM fails. I like it that the authors dig deep into this and seek for explanation, but the evidence provided looks more like correlations than explanation. I would encourage authors to reorganize to make this part more of motivation for the proposed method instead of challenging previous methods. 2) For eq (4), what happens if I choose \eta ~ U([-\beta, beta]^d). Will \beta appear in the bound? And if so, do you want to weaken the claim in this section? 3) I would be interested to see one learning curve with iterations/epochs as x-axis and robust accuracy as y-axis, and show when does the catastrophic overfitting happen for different methods, maybe train CIFAR-10 with \epsilon=14 for 60 epochs . 4) I would also like to explicitly see the effect of the proposed regularizer on the cosine distance in Figure 4. 5) Large scale dataset like ImageNet will be a plus. What happens when we use \epsilon > 4 for ImageNet, Do all methods including the proposed one fail? after rebuttal: I will keep the previous score. I still have concerns about the practical usage and the theoretical analysis, but this paper looks better than a lot of other papers I reviewed this year.


Review 4

Summary and Contributions: Starting from the catastrophic overfitting phenomenon of FGSM based adversarial training, this paper investigated the underlying reasons and possible ways to improve it. Based on the analysis, a GradAlign method is proposed to improve FGSM based adversarial training.

Strengths: The paper is well motivated and structured. Starting from the catastrophic overfitting phenomenon, it first revisited the existing random initialization based explanation and its limitations. Then it further introduces the gradient alignment perspective. The analysis is interesting and is presented very clear by leveraging both simple examples and theoretical analyses.

Weaknesses: The extent to which this analysis holds is not analyzed in the paper. The empirical analysis is mainly done on CIFAR10 only.

Correctness: The proposed method is correct.

Clarity: The paper is well structured and the presentation is clear.

Relation to Prior Work: Relation to prior work is clearly discussed and the distinctions from previous works are clear.

Reproducibility: Yes

Additional Feedback: (1) It is mentioned that catastrophic overfitting does not occur for the experiments on ImageNet. What is the potential reason? How does the gradient-alignment value evolve in this case? (2) This work tries to answer the question of when and why FGSM adversarial training works. However, the when question is investigated to a limited extent. For example, currently the empirical investigations are mainly limited to the domain of low-resolution images. =======post rebuttal update The major concern on the extent to which this analysis holds is still not clear. The ImageNet result mentioned in the paper is not helpful in understanding this. I do encourage the authors to make some attempts to include some additional data points if possible.

[Author Response · NeurIPS 2020]

We thank all reviewers for the encouraging feedback and detailed comments which we'll integrate into the next version.

**R1, R2, R3**: *FGSM+GradAlign is slower than FGSM.*
We admit that `GradAlign` leads to a slowdown: e.g. on CIFAR-10 the runtime of FGSM AT is 9.5 min while FGSM
+ `GradAlign` AT takes 30.9 min on an NVIDIA V100 GPU. However, we present `GradAlign` as a proof of concept
motivated by our empirical and theoretical analysis. We hope that the same effect (stability of the gradients under
*random* noise) can be achieved in future work with other regularization methods that avoid double backpropagation.

**R1**: *What makes the grad. alignment low by FGSM training? Why a noise-sensitive filter is learned by FGSM training?*
These are very interesting questions, and we believe they are connected to the finding of [25] that SGD for neural
networks learns models of increasing complexity, e.g., measured in terms of local linearity.

**R2**: *Running stronger attacks, e.g., AutoAttack or MultiTargeted*
After running *AutoAttack*, we observe that it proportionally reduces the adversarial accuracy for all methods. E.g., for
$\varepsilon = 8/255$, FGSM+`GradAlign` achieves 44.54±0.24% adversarial accuracy while FGSM-RS achieves 42.80±0.58%.
This is consistent with the results of the *AutoAttack* paper where they show an average reduction of 2%–3% adverarial
accuracy compared to most of the evaluations that were originally done with variants of PGD.

**R2**: *Discussion on LLR [26] and CURE [24]. Their exact goal is to make the loss surface smoother.*
Indeed, the goals of LLR/CURE and `GradAlign` wrt smoothness are similar, however the important difference is that
we were not looking for a replacement of adv. training, but rather for a *complement* that would prevent catastrophic
overfitting. Related to this, in Table 7 we also provide the results for FGSM+CURE where we can see that CURE
also stabilizes FGSM training, but performs worse than FGSM+`GradAlign`. Finally, `GradAlign` does not have any
worst-case motivation unlike CURE (uses the FGSM point) or LLR (uses a point with the worst-case linear violation).

**R2**: *Line 118: How is the alpha step-size tuned for this experiments? (is it $1.25\varepsilon$?)*
Yes, we used $\alpha = 1.25\varepsilon$ since it was the recommended choice of Wong et al. [44] on all the datasets they considered.

**R2**: *Performance of FGSM+GradAlign in large $\varepsilon$ settings (16/255) on ImageNet against random targeted attacks.*
First, we note that for FGSM+`GradAlign` on CIFAR-10 we did not encounter cat. overfitting even with $\varepsilon = 16/255$. We
briefly tried *targeted* AT with $\varepsilon = 16/255$ on ImageNet but we did not succeed at training a sufficiently robust model. We
think that it is likely that more epochs (e.g., LLR [26] used 110 epochs instead of 15 epochs as we did following [44])
and different hyperparameters are needed, but tuning them on ImageNet was too computationally expensive for us.

**R3**: *Learning curve with robust accuracy and the effect of the regularizer*
*on grad. alignment, e.g., on CIFAR-10 with $\varepsilon = 14$ and 60 epochs.*
We present this experiment in Fig. A. The only two methods that do not fail
at epoch 60 are PGD-10 and FGSM+`GradAlign` which is also reflected
by their gradient alignment, i.e. cosine distances (highest for `GradAlign`).

**R3**: *I'd vote for early stop since it is not complicated and rather efficient.*
We'd like to clarify that using early stopping leads to worse PGD accuracy
(particularly for high $\varepsilon$) and much worse clean accuracy as we comment
in lines 307-310. Thus, early stopping alone is not a satisfying solution.

**R3**: *What happens when we use $\varepsilon > 4$ for ImageNet? Do all methods*
*including the proposed one fail?*
We have the results for $\varepsilon = 6$ in Table 6 and catastrophic overfitting there
occurs *only* for the FGSM-RS model. However, the results on ImageNet
are not fully conclusive since we could not repeat the experiments over
multiple random seeds due to the computational constraints.

**R3**: *Lemma 1: what if $\eta \sim \mathcal{U}([-\beta, \beta]^d)$? Will $\beta$ appear in the bound?*
We were interested in $\beta = \varepsilon$ since it was the setting of [44]. Indeed, $\beta$ will
appear in the bound, but this will not change the message of Lemma 1.

**R4**: *The empirical analysis is mainly done on CIFAR10 only.*
We'd like to emphasize that we have provided experiments on ImageNet
to illustrate that `GradAlign` can be scaled to large datasets. But due to our

Figure A: Illustration of catastrophic overfitting for various AT methods with $\varepsilon = 14/255$.

limited computational resources, we could not do replications over random seeds and a thorough comparison to other
methods (particularly, to PGD-10) for a range of $\varepsilon$ as on CIFAR-10 (e.g., Fig. 1 required to train *480 different models*).

**R4**: *Why does cat. overfitting not occur on ImageNet [for $\varepsilon \in \{2, 4\}$]? How does the gradient alignment evolve?*
The main reason is that these $\varepsilon$ values are sufficiently small (we observed the same also on CIFAR/SVHN for $\varepsilon \leq 4/255$).
The gradient alignment decreases gradually over epochs, but *without* a sharp drop that would indicate cat. overfitting.

[Meta-Review · NeurIPS 2020]

The reviewrs reached the concensus of accepting this paper. All the reviewers think this paper provides an interesting analysis of adversarial training and could inspire future work in the community. The experimental results are somehow weaker (e.g., lack of ImageNet experiments), and we hope the authors can further improve the paper based on review comments.